# Semantic Integrity Matters: Benchmarking and Preserving High-Density Reasoning in KV Cache Compression

**Xiang Liu** [1]   **Zhenheng Tang** [2]   **Hong Chen** [1]   **Peijie Dong** [1]   **Zeyu Li** [1]   **Xiuze Zhou** [1]   **Bo Li** [2 3]   **Xuming Hu**[✉ 1]   **Xiaowen Chu**[✉ 1]

## Abstract

While Key-Value (KV) cache compression is essential for efficient LLM inference, current evaluations disproportionately focus on **retrieval-oriented** long-context tasks, potentially masking the degradation of High-Density Reasoning where Chain-of-Thought (CoT) coherence is critical. We introduce KVFundaBench to systematically evaluate this gap, revealing a sharp dichotomy: while retrieval tasks remain robust, reasoning tasks exhibit severe Task-Dependent Degradation under aggressive compression due to disrupted CoT links. Extending our analysis to the DeepSeek-R1 model, we uncover that its specialized attention patterns offer unique insights into the fragility of reasoning chains. Guided by these findings—specifically the necessity of preserving few-shot examples as indivisible **Semantic Units**—we propose ShotKV. This approach explicitly separates prefill and decoding phases to prioritize semantic integrity. Empirical results demonstrate that ShotKV achieves 9%-18% accuracy improvements on long-context generation tasks and effectively generalizes to document QA, all while delivering an 11% latency reduction compared to full cache inference.

## 1. Introduction

The evolution of Large Language Models (LLMs) to process large documents for tasks such as answering and summarizing questions (Raffel et al., 2020; Chowdhery et al., 2023; Yu et al., 2026; Touvron et al., 2023a;b; Tang et al., 2026),

[1]The Hong Kong University of Science and Technology (Guangzhou), Email: xliu886@connect.hkust-gz.edu.cn [2]The Hong Kong University of Science and Technology [3]Guangzhou HKUST Fok Ying Tung Research Institute. Correspondence to: Xuming Hu <xuminghu@hkust-gz.edu.cn>, Xiaowen Chu <xwchu@hkust-gz.edu.cn>.

*Proceedings of the 43rd International Conference on Machine Learning*, Seoul, South Korea. PMLR 306, 2026. Copyright 2026 by the author(s).

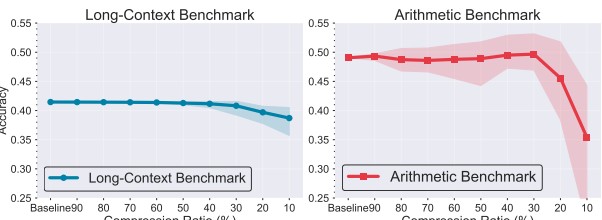

*(a)* KV cache compression methods on long-context and arithmetic benchmarks.

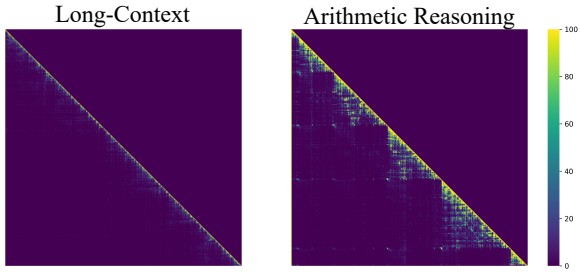

*(b)* Attention heatmap.

*Figure 1.* KV cache compression methods on long-context and arithmetic benchmarks. (a) Arithmetic benchmark shows more performance degradation than long-context benchmark. (b) Long-Context benchmark shows more sparsity in attention heatmap.

spurred by breakthroughs in system architectures (Dao et al., 2022; Dao, 2024; Jacobs et al., 2023; Xiao et al., 2024; Zhang et al., 2023a; Huang et al., 2026) and model design (Chen et al., 2023a; Xiong et al., 2024; Chen et al., 2023b; Peng et al., 2024), has significantly increased GPU memory demands during inference (AI21, 2024; X.AI, 2024; Reid et al., 2024; Anthropic, 2024; DeepSeek-AI, 2024; Liu et al., 2024a), making the development of efficient key value (KV) cache compression strategies a critical focus for LLM deployment and optimization. This concern is shared across the broader efficiency stack, including memory-efficient fine-tuning (Pan et al., 2024), token-efficient adaptive inference (Liu et al., 2026), and reasoning-model inference serving (Li et al., 2026); KV-level intervention is the axis we focus on here.

To address this, numerous studies have proposed selective token retention strategies (Xiao et al., 2024; Zhang et al., 2023b; Li et al., 2024b; Ge et al., 2024; Cai et al., 2024; Fu et al., 2024; Yang et al., 2024; Adnan et al.,

2024; Liu et al., 2023; Tang et al., 2024), with pioneering works such as H2O (Zhang et al., 2023b) and SnapKV (Li et al., 2024b) showing that retaining approximately 50% of KV cache entries can balance model performance with significant memory savings. However, current research predominantly evaluates these methods on retrieval-oriented long-context benchmarks, where the dominant difficulty is locating salient spans in a long context—e.g., Needle-In-A-Haystack (NIAH) (Kamradt, 2023) and many tasks in LongBench (Bai et al., 2024; 2025). We argue that this evaluation focus can mask the true fragility of compressed models on a different regime, which we term **High-Density Reasoning**: tasks where nearly every token in a multi-step prompt (e.g., chain-of-thought few-shot exemplars) is reasoning-critical, in contrast to retrieval tasks where only a small "needle" subset truly matters. As illustrated in Figure 1(a), under matched compression methods, an arithmetic-reasoning workload exhibits sharp degradation while a retrieval-oriented long-context workload remains comparatively stable; the corresponding attention heatmap in Figure 1(b) shows the contrasting sparsity patterns that motivate this study. Under matched settings, many-shot reasoning at moderate context lengths can be *more* compression-sensitive than retrieval over much longer contexts, because breaking a single semantic link in the Chain-of-Thought (CoT) can lead to catastrophic reasoning failure. Consequently, the impact of compression on a spectrum of fundamental LLM capabilities—such as *arithmetic reasoning*, *world knowledge*, *commonsense reasoning*, and *safety*—remains largely unexplored, particularly concerning their distinct attention patterns.

To this end, we introduce **KVFundaBench**, a benchmark designed to systematically assess the effects of KV cache compression across these diverse fundamental capabilities and their underlying attention dynamics. The benchmark includes 5 categories of tasks: *world knowledge*, *commonsense reasoning*, *arithmetic reasoning*, *code generation*, and *safety*. Utilizing KVFundaBench, we conduct an extensive evaluation of **six representative KV cache compression methods** across **four state-of-the-art LLMs**, including both standard instruction-tuned models and specialized reasoning models. Our KVFundaBench reveals several key findings: (1) *Task-Dependent Degradation*: Performance degradation is highly task-dependent, with arithmetic reasoning tasks showing particular sensitivity to aggressive compression; (2) *Model-Type Robustness*: Multi-step reasoning LLMs demonstrate higher compression robustness compared to instruction-tuned models; (3) *Prompt Length Vulnerability*: Shorter prompts are more vulnerable to compression effects; (4) *Chunk-Level Superiority*: Chunk-level compression strategies show superior performance on complex long-context reasoning tasks; (5) *Prompt-Gain Sensitivity*: Tasks with larger prompt-based performance

gains exhibit higher compression sensitivity; and (6) *Long-Context Generation Sensitivity*: Long-context generation tasks are particularly sensitive to compression. These findings motivate **ShotKV**, a lightweight, *insight-driven proof-of-concept* that operationalizes the central conclusion of KVFundaBench: preserving few-shot examples as indivisible **Semantic Units**. ShotKV is intentionally simple—a task-motivated combination of shot-aware prefill preservation and dynamic decoding compression—and its purpose is to validate, not to claim algorithmic novelty over, the benchmark's hypothesis that semantic-unit preservation is key to robustness under aggressive compression.

Our main contributions are summarized as follows:

- We introduce **KVFundaBench** to systematically evaluate the effects of KV cache compression on **High-Density Reasoning** and fundamental capabilities.
- Our systematic investigation reveals multiple critical factors influencing compression sensitivity, including model training dynamics, prompt length characteristics, task-specific requirements, long-context reasoning, and long-context generation capabilities.
- We propose ShotKV, a semantic-aware strategy that treats few-shot examples as indivisible **Semantic Units**. By explicitly separating prefill and decoding compression, ShotKV achieves **9%-18%** accuracy gains on long-context generation and generalizes to document QA, while delivering an **11% reduction in end-to-end latency**

**Conflict of Interest Disclosure.** All authors are affiliated with academic institutions; financial support is fully disclosed in the Acknowledgments. The authors declare no additional competing financial or non-financial interests that could be perceived to influence this work.

## 2. Related Work

**KV cache compression.** A growing line of work reduces KV memory by selecting a subset of tokens to retain. Token-level methods such as StreamingLLM (Xiao et al., 2024), H2O (Zhang et al., 2023b), SnapKV (Li et al., 2024b), FastGen (Ge et al., 2024), Scissorhands (Liu et al., 2023), Quest (Tang et al., 2024), Keyformer (Adnan et al., 2024), and value-aware variants (Guo et al., 2024) score individual tokens via attention statistics and evict the rest. Layer- or budget-aware variants such as PyramidKV (Cai et al., 2024), PyramidInfer (Yang et al., 2024), and Ada-KV (Feng et al., 2025) adapt the budget across layers or heads, while cross-layer designs (CLA (Brandon et al., 2024), MiniCache (Liu et al., 2024b), YOCO (Sun et al., 2024), LCKV (Wu & Tu, 2024)) share or merge KV caches across layers. Most relevant to us, ChunkKV (Liu et al., 2025b) preserves contiguous chunks rather than isolated tokens, and SCOPE (Wu et al., 2025) separates prefill and decoding compression. Recent work also benchmarks compression breadth (Yuan

et al., 2024; Devoto et al., 2025; Kim et al., 2025; Liu et al., 2025a; Zhu et al., 2025; Chen et al., 2026), asks which capabilities should be preserved under compression (Tang et al., 2025; Dong et al., 2025), and reuses caches for RAG (Yao et al., 2025). Orthogonal to token retention, KV *quantization* (Li et al., 2025) reduces memory by lower-precision representations and can be combined with retention-based methods.

**Long-context evaluation.** Existing long-context benchmarks predominantly emphasize *retrieval-oriented* settings: NIAH (Kamradt, 2023), RULER (Hsieh et al., 2024), ∞-Bench (Zhang et al., 2024a), and many tasks in Long-Bench (Bai et al., 2024; 2025) measure how well a model locates salient spans within long contexts. Generation-oriented evaluation (Liu et al., 2024d) and many-shot ICL (Agarwal et al., 2024) are more recent and remain comparatively underexplored under compression. KVFundaBench complements these by systematically probing fundamental capabilities (Hendrycks et al., 2021a; Suzgun et al., 2023; Cobbe et al., 2021; Talmor et al., 2019; Chen et al., 2021; Lin et al., 2022; Hartvigsen et al., 2022) under matched compression budgets, and ShotKV positions itself as a shot-aware combination of (i) chunk-style semantic-unit preservation in prefill and (ii) dynamic decoding compression—designed not as a new algorithmic primitive, but as a targeted instantiation of the insight that emerges from KVFundaBench. An extended discussion is provided in Section C.

# 3. Preliminary

In this section, we provide comprehensive preliminaries of KV cache compression and LLM evaluation.

**Key-Value Cache in LLMs** With the increasing long-context capabilities of LLMs, the Key-Value (KV) cache has become crucial for improving inference efficiency. During LLM inference, the KV cache stores intermediate computation results to avoid redundant calculations. For a given input sequence $x = (x_1, x_2, ..., x_n)$, each transformer layer $l$ maintains its key cache $K^l = (k_1^l, k_2^l, ..., k_n^l)$ and value cache $V^l = (v_1^l, v_2^l, ..., v_n^l)$, where $k_i^l, v_i^l \in \mathbb{R}^d$ represent the key and value vectors for token $x_i$ at layer $l$.

**KV Cache Compression** KV cache compression aims to reduce memory usage by selectively storing or merging cached vectors. A compression operation can be denoted as $C(K, V) = (K', V')$, where $K'$ and $V'$ are compressed caches with size $m < n$, where $C$ is the compression method, $m$ is the number of retained tokens, and $n$ is the original number of tokens. The core idea is token selection - identifying and retaining important tokens based on attention patterns or other metrics while discarding less important ones. The compression ratio $r = m/n$ indicates how aggressively the cache is compressed, where a smaller

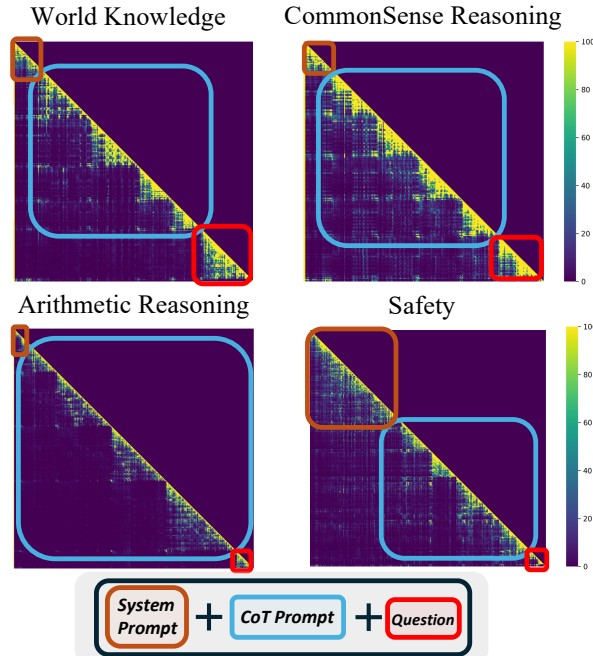

*Figure 2.* Attention heatmap on different tasks.

ratio means more aggressive compression.

**Evaluation Protocol** To thoroughly evaluate the impact of KV cache compression on LLMs' capabilities, we assess five benchmark categories: world knowledge, commonsense reasoning, arithmetic reasoning, code generation, and safety.

For each task category and compression method $C$, we calculate the relative performance change as follows:

$$\Delta P = \frac{P_C - P_{\text{base}}}{P_{\text{base}}} \quad (1)$$

where $P_C$ and $P_{\text{base}}$ represent the performance scores with and without compression, respectively.

# 4. Benchmark Design

### 4.1. Benchmark Setups

In this section, we will introduce the KVFundaBench setups, including the datasets, models, and evaluation environment.

**Datasets** To evaluate the performance of KV cache compression on LLMs' overarching capabilities, we assess five benchmark categories: *World Knowledge (WK)* using MMLU (Hendrycks et al., 2021a), measured by accuracy; *CommonSense Reasoning (CSR)* using CommonsenseQA (Talmor et al., 2019), evaluated through multiple-choice accuracy; *Arithmetic Reasoning (AR)* using GSM8K (Cobbe et al., 2021), assessed by solve rate; *Code Generation (CG)* using HumanEval (Chen et al., 2021), measured by pass@1 rate on test cases; and *Safety (SA)* using JailBreakV (Luo et al., 2024), evaluated by attack success rate. Furthermore, we test the performance of KV cache compression on LongGenBench (Liu et al.,

2024d), a ***long-context generation (LG)*** benchmark. Detailed statistics for all datasets are provided in Section F.1.

**Models** We conduct experiments on a series of LLMs, including LLaMA-3.1-8B, LLaMA-3.1-8B-Instruct (Dubey et al., 2024), Mistral-7B-Instruct (Jiang et al., 2023a), and multi-step reasoning LLM DeepSeek-R1-Distill-Llama-8B (DeepSeek-AI et al., 2025).

**KV Cache Compression Methods** To thoroughly investigate the potential impact on KV cache compression methods, we select the following methods: StreamingLLM (Xiao et al., 2024), SnapKV (Li et al., 2024b), H2O (Zhang et al., 2023b), PyramidKV (Cai et al., 2024), PyramidInfer (Yang et al., 2024), and ChunkKV (Liu et al., 2025b).

**Hyperparameters** The hyper-parameters for different observations are shown in Table 18. The temperature for the experiments are set to 0 for ensuring the deterministic results.

**Evaluation Environment** We use the lm-evaluation-harness (Gao et al., 2023) library to load the models and evaluate the performance. The evaluation environment is a NVIDIA A40 GPU server.

### 4.2. Attention Pattern Analysis on KVFundaBench

To better understand the task-specific sensitivity to KV cache compression observed in our experiments, we analyze the attention patterns of LLMs on KVFundaBench. This analysis provides key insights into why different tasks respond differently to compression, from two perspectives: (1) task-specific attention patterns; and (2) attention distributions.

As shown in Figure 2, our benchmark reveals different attention behaviors in tasks: world knowledge and common sense reasoning tasks exhibit **universal attention distributions**, while arithmetic reasoning and safety tasks display more **specialized patterns**. In particular, arithmetic reasoning tasks show increased attention sparsity (i.e. focused attention on individual prompt examples), and safety tasks concentrate attention on system prompts, in contrast to the more uniform attention distribution seen in world knowledge and commonsense reasoning.

To further investigate the attention dynamics that might explain the task-specific sensitivities to KV cache compression, we analyzed the cumulative attention score distributions, as illustrated in Figure 3. Figure 3(a) depicts the overall attention distribution, which includes the initial sink tokens (Xiao et al., 2024). In this view, both long-context and arithmetic tasks demonstrate a very similar pattern: a steep initial rise where the top 1% of tokens capture over 60% of the total attention mass. This highlights the predominant role of sink tokens in attracting attention, regardless of the specific task.

However, a more distinct pattern emerges when these initial sink tokens (specifically, the first four tokens) are ex-

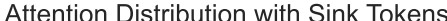

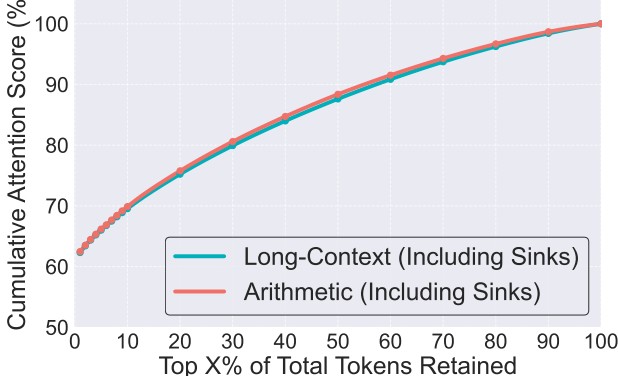

*(a)* Attention Distribution with Sink Tokens

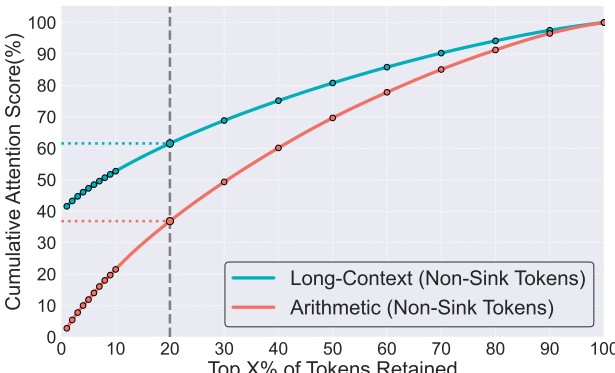

*(b)* Attention Distribution without Sink Tokens

*Figure 3.* Cumulative attention score distribution for Long-Context and Arithmetic. (a) Overall distribution including initial sink tokens, showing high initial concentration. (b) Distribution without sink tokens (first 4 tokens removed), revealing that Arithmetic's non-sink attention is more diffuse compared to Long-Context's.

cluded from the analysis, as shown in Figure 3(b). Within the remaining non-sink tokens, the attention distribution for arithmetic tasks becomes notably more diffuse, with a slower accumulation of attention mass. This divergence suggests that while sink tokens provide a common, strong attentional anchor, the subsequent distribution of attention across task-relevant (non-sink) tokens varies. The more diffuse attention in arithmetic's non-sink tokens implies a reliance on a broader set of contextual cues for its structured reasoning, potentially making it more vulnerable when compression begins to impact these non-sink tokens.

These detailed analyses of attention distributions (Figure 2 and Figure 3) reveal that LLMs engage different contextual information and attention strategies when performing long-context tasks versus tasks requiring fundamental abilities such as arithmetic reasoning. This highlights the necessity of evaluating KV cache compression beyond long-range dependencies to specifically assess its impact on diverse fundamental capabilities, owing to their distinct attentional mechanisms.

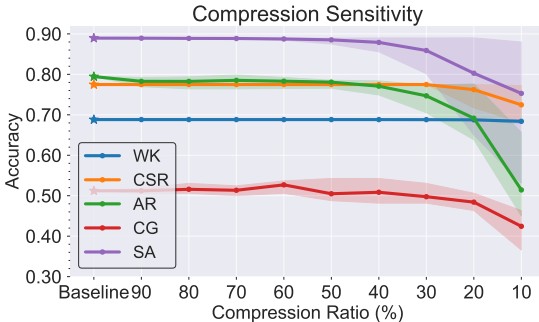

*(a)* Sensitivity Analysis of Different Benchmark Categories to KV Cache Compression

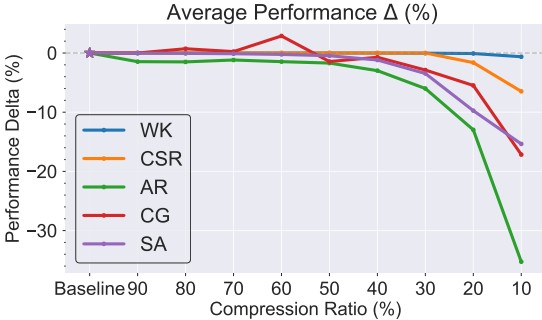

*(b)* Performance Delta Lines with Baseline

*Figure 4.* Sensitivity Analysis of Different Benchmark Categories to KV Cache Compression. The benchmark categories include **WK** (World Knowledge), **CSR** (CommonSense Reasoning), **AR** (Arithmetic Reasoning), **CG** (Code Generation), and **SA** (Safety). The performance delta lines are calculated by Equation (1).

### 4.3. Results and Analysis

In this section, we present the results and an analysis of the experiments. For detailed results, see Section D.1.

***Observation 1.*** **Task-Dependent Degradation**: KV cache compression methods show task-dependent performance degradation, WK and CSR are more robust to KV cache compression. As demonstrated in Figure 4, all tasks maintain stable performance at compression ratios above 40%, but exhibit distinct degradation patterns below this threshold. *Arithmetic reasoning*, *code generation*, and *safety* tasks demonstrate the highest compression sensitivity, characterized by precipitous performance declines. Figure 5 illustrates the detailed performance impact of various KV cache compression methods across different tasks. This degradation is most pronounced in *arithmetic reasoning* (Figure 5(c)), where performance deteriorates significantly below the compression ratio of 20%, with accuracy dropping from approximately 0.75 to below 0.5. Among the evaluated methods, ChunkKV (Liu et al., 2025b) and PyramidKV (Cai et al., 2024) consistently demonstrate superior stability in most tasks, while StreamingLLM (Xiao et al., 2024) exhibits increased sensitivity to aggressive compression. Additionally, *R1-Arithmetic reasoning* (Figure 5(f)) indicates that reasoning LLMs demonstrate enhanced ro-

bustness to KV cache compression. Highlighting *World Knowledge* and *Common Sense Reasoning* as the most robust tasks, indicating that these tasks are less sensitive to KV cache compression.

***Observation 2.*** **Model-Type Robustness**: Multi-step reasoning LLMs are more robust to KV cache compression. Figure 6 presents a comparative analysis of LLaMA-3.1-8B across its base (w/o instruct tuned), instruct-tuned, and DeepSeek-R1 distilled variants, illustrating their averaged performance in five compression methods with confidence intervals. The R1 distilled model demonstrates superior stability, maintaining performance around 0.60 even at a compression ratio 10%. The instruct-tuned model achieves a higher initial accuracy (0.8), but it exhibits heightened compression sensitivity, with performance deterioration beginning at 30% compression ratio and declining sharply to approximately 0.5 at 10% ratio. These findings suggest that while multi-step reasoning LLMs demonstrate enhanced robustness to KV cache compression, and instruct-tuning improves overall model performance, the latter may inadvertently increase model vulnerability to aggressive compression, particularly at compression ratios below 30%.

***Observation 3.*** **Prompt Length Vulnerability**: Shorter prompts are more vulnerable to KV cache compression. As illustrated in Figure 7, the effect of KV cache compression is markedly different with varying prompt lengths (shot numbers). Scenarios with fewer shots (for example, one-shot and two-shot) demonstrate heightened sensitivity to compression; their performance degrades more precipitously below a compression ratio of 30% compared to scenarios with a greater number of shots (e.g., 4-8 shots). For example, in 1-shot settings, performance decreases from 0.5 to 0.05 as the compression ratio decreases from 30% to 10%. In contrast, 8-shot settings experience a less severe reduction, from 0.75 to 0.5, under the same compression conditions. This suggests that prompts with more shots, by virtue of containing more contextual examples, offer a richer set of reference points for the model. Consequently, the model's reliance on any single example being perfectly preserved in the compressed KV cache is reduced, leading to greater robustness against aggressive compression.

***Observation 4.*** **Chunk-Level Superiority**: Chunk-level compression is more effective for long-context structured reasoning tasks. Inspired by Agarwal et al. (2024), we consider many-shot in-context learning as a long-context reasoning task, which is more complex than existing long-context benchmarks, such as LongBench and NIAH. Figure 8 shows the performance of KV cache compression methods on a 50-shot GSM8K task, where the prompt length exceeds 4K tokens. From the figure, we observe that ChunkKV (Liu et al., 2025b) demonstrates the most stability when the compression ratio is below 10% on both LLaMA-3.1-8B-Instruct and DeepSeek-R1-Distill-Llama-8B, indicating that in more

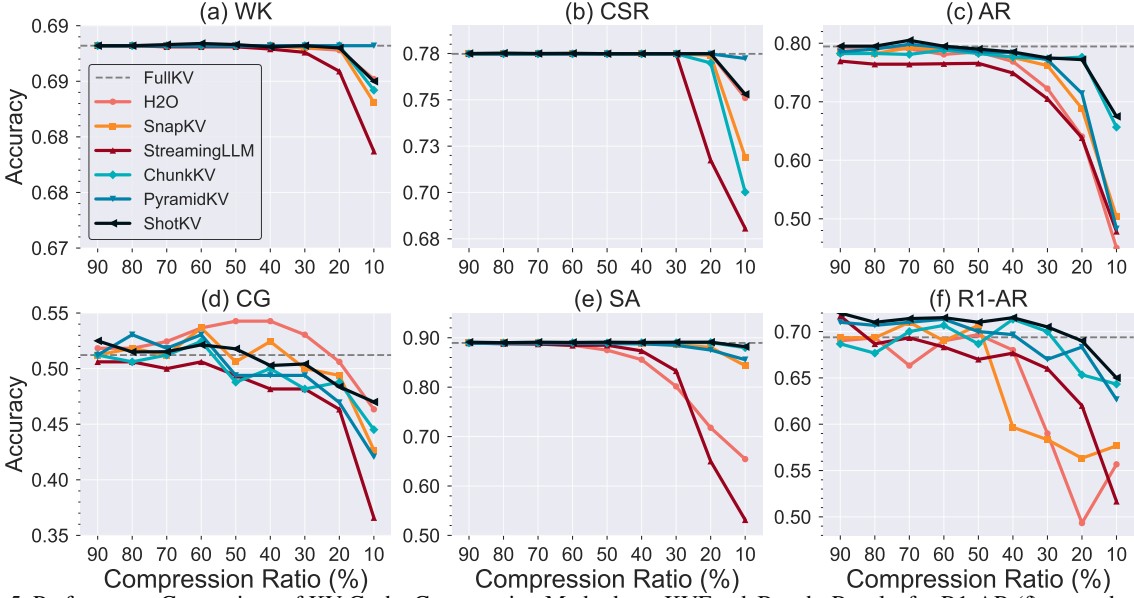

*Figure 5.* Performance Comparison of KV Cache Compression Methods on KVFundaBench. Results for R1-AR (f) were obtained using the DeepSeek-R1-Distill-Llama-8B model. ShotKV is our proposed method; details can be found in Section 5.

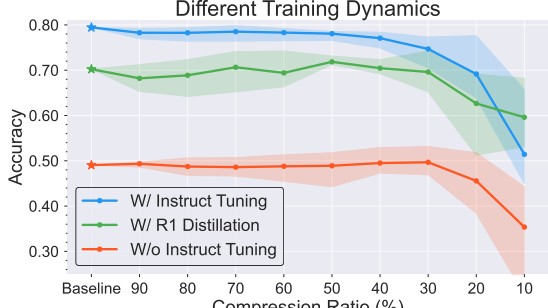

*Figure 6.* Performance Comparison of KV Cache Compression Methods on different training dynamics on *Arithmetic Reasoning*

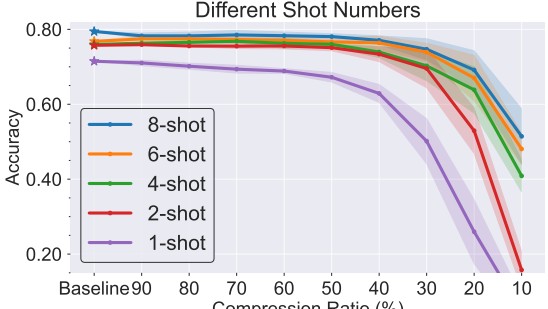

*Figure 7.* Average Performance Across Different Shot Numbers

complex long-context arithmetic reasoning tasks, chunk-level retention is more effective at preserving semantic information. This highlights the effectiveness of chunk-level compression for long-context structured reasoning tasks.

***Observation 5.*** **Prompt-Gain Sensitivity**: Tasks with higher performance gains from ICL and CoT are disproportionately more sensitive to KV cache compression. As shown in Table 1, different tasks exhibit varying levels of performance improvement when moving from zero-shot to CoT prompting. *Arithmetic reasoning* shows a dramatic improvement of 50.41%, whereas *World Knowledge* demon-

*Table 1.* Zero-shot vs Few-shot Performance Comparison

| Benchmark | Zero-shot ↑ | CoT ↑ | Delta∆ |
|---|---|---|---|
| *Arithmetic Reasoning* | 29.04 | 79.45 | +50.41 |
| *World Knowledge* | 62.62 | 68.82 | +6.20 |

*Table 2.* KV cache compression methods' performance on *LG-GSM8K*. Comp. % indicates the compression ratio.

| Method & Comp. % | 100% | 40% | 35% | 30% | 25% |
|---|---|---|---|---|---|
| FullKV | 46.00 | - | - | - | - |
| StreamingLLM | - | 39.50 | 28.67 | 14.83 | 6.33 |
| H2O | - | 32.66 | 25.17 | 19.83 | 14.83 |
| PyramidInfer | - | 38.33 | 27.67 | 20.50 | 16.67 |
| ShotKV(Ours) | - | **47.33** | **41.33** | **38.33** | **26.83** |

strates a much more modest gain of 6.20%.

From Figure 4, we observe a strong correlation between prompt-based performance gains and compression sensitivity. Tasks that benefit significantly from CoT prompting, such as *Arithmetic Reasoning*, are markedly more vulnerable to KV cache compression. This suggests a high dependency on the precise preservation of prompt information: when a model relies heavily on the reasoning patterns provided in the prompt to boost performance, any loss of this information via compression leads to a disproportionate drop in accuracy. Conversely, tasks like *World Knowledge*, which show modest gains from CoT, exhibit greater resilience, likely because the model draws primarily from its internal parametric knowledge rather than the specific examples in the context.

***Observation 6.*** **Long-Context Generation Sensitivity**: KV cache compression exhibits significant performance degradation in long-context generation tasks. As demonstrated in Table 2, our evaluation of three unified com-

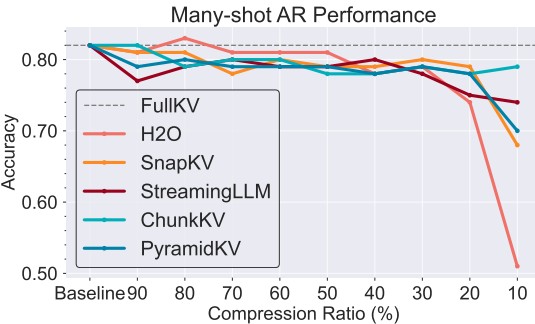

*(a)* Many-shot *Arithmetic Reasoning* on LLaMA3.1-8B-Instruct

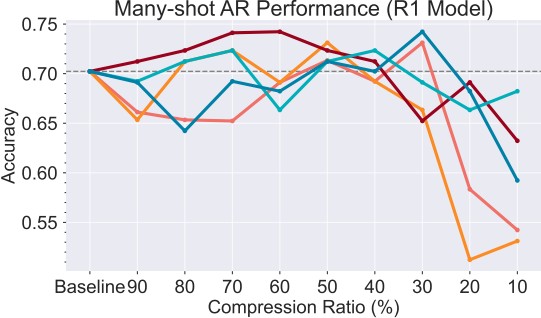

*(b)* Many-shot *Arithmetic Reasoning* on DeepSeek-R1-Distill-Llama-8B

*Figure 8.* Many-shot scenario on KV cache compression

pression methods—StreamingLLM, H2O, and PyramidInfer—on *LG-GSM8K* reveals substantial performance limitations. **We exclude context-compression methods like ChunkKV and SnapKV from this setting, as they primarily target static prefill compression and lack the dynamic token eviction mechanisms required to manage the rapidly growing cache during extended generation phases (e.g., 4k+ tokens).** In this arithmetic reasoning task with approximately 4k token generation duration, the evaluated unified compression methods show notable deterioration, with performance declining by more than 20% at compression ratios below 30%. The ShotKV is our proposed method that aims to improve the performance of KV cache compression on Long-Context Generation tasks, details in Section 5.

**Summary and Design Implications** Our comprehensive analysis reveals distinct challenges for KV cache compression: (1) The structural integrity of few-shot examples is crucial for reasoning tasks (*Obs. 4*), suggesting that compression should operate at the *semantic unit (shot) level* rather than the token level. (2) The conflicting requirements of preserving static prompt instructions (*Obs. 1 & 5*) versus dynamic generation contexts (*Obs. 6*) indicate that a unified compression strategy is suboptimal. These insights motivate the design of ShotKV, which explicitly separates prefill and decoding compression strategies and incorporates a shot-aware selection mechanism to preserve semantic coherence.

## 5. ShotKV

Based on the insights from KVFundaBench, we identify two critical design principles for effective KV cache compression on fundamental abilities:

1. **Semantic Completeness (Addressing Obs. 4 & 5):** Since reasoning tasks rely on coherent logical chains found in few-shot examples, compression algorithms must preserve these shots as atomic units. Token-level dropping (e.g., H2O) fragments these units, leading to performance collapse in arithmetic reasoning.

2. **Phase Separation (Addressing Obs. 6):** The prefill phase (encoding static knowledge/rules) and decoding phase (generating dynamic steps) exhibit different sensitivities. A unified strategy fails to balance these distinct needs.

We emphasize that ShotKV is intentionally simple: it is a *task-motivated combination* of two design choices derived from KVFundaBench—(i) shot-aware semantic-unit preservation in prefill, sharpening the chunk-level retention insight of ChunkKV with explicit few-shot boundaries, and (ii) explicit prefill/decoding separation, addressing the dynamic information needs identified in Observation 6. We do not claim novelty for generic attention-based ranking; rather, our contribution is to show that this targeted, lightweight combination is sufficient to recover the reasoning performance lost by token-level methods, thereby validating the central conclusion of KVFundaBench: preserving *semantic units* matters more than fine-grained token selection under aggressive budgets.

### 5.1. Implementation

The **ShotKV** (Prefill-Decoding Separated **Shot**-aware **KV** Cache Compression), which separates the compression strategy for prefill and decoding phases. The key insight is that the prefill phase KV cache, which contains crucial prompt information, should be compressed once and remain fixed, while the decoding phase KV cache can be dynamically compressed with different strategies.

Given a prompt with $n$ shots and tokens generated, we define:
$$KV_{\text{total}} = KV_{\text{prefill}} \cup KV_{\text{decoding}} \quad (2)$$

For the prefill phase, guided by **Observation 4** (Chunk-Level Superiority) which highlights the necessity of maintaining semantic chunks, we compute shot importance and preserve complete shot examples:
$$\text{Score}_{\text{prefill}}^l(s_i) = \frac{1}{k_i} \sum_{t \in s_i} \sum_{h=1}^{H} \alpha_{t,h}^l \quad (3)$$

where $s_i$ represents the $i$-th shot example containing $k_i$ tokens. The term $\alpha_{t,h}^l$ denotes the accumulated attention scores of token $t$ within shot $s_i$ in attention head $h$ at transformer layer $l$. Following standard practices in KV compression (Zhang et al., 2023b; Li et al., 2024b), we treat

the accumulated attention weight as a *practical heuristic* for ranking semantic units rather than a principled measure of semantic importance: it is empirically effective at preserving coherent few-shot structure under a budget, and we report sensitivity to shot ordering and exemplar choice in the appendix. The compression is performed independently for each layer, allowing different layers to retain different subsets of shots based on their specific attention patterns. Once the prefill phase KV cache is compressed, it remains fixed throughout the generation process.

Given a prefill compression ratio $r_p$, we prioritize shots with higher scores while ensuring that the total number of preserved tokens does not exceed the KV cache limit. Specifically, the shots are ranked by their scores and selected in descending order until they reach the compression budget $r_p \times |KV_{\text{prefill}}|$. This shot-level selection strategy helps to maintain the semantic coherence of important examples while adhering to memory constraints.

$$KV_{\text{prefill},l}^{C} = \text{Compress}(\{s_i | s_i \in S_{\text{preserved},l}^{*}\}) \quad (4)$$

$$S_{\text{preserved},l}^{*} = \text{TopK}\left(\{s_1, \ldots, s_n\}, \text{Score}_{\text{prefill}}^{l}, N_{\text{budget}}\right) \quad (5)$$

$$\text{subject to:} \quad \sum_{s_i \in S} k_i \leq r_p \times |KV_{\text{prefill}}| \quad (6)$$

Here, $KV_{\text{prefill},l}^{C}$ represents the compressed KV cache for prefilling at layer $l$, and $S_{\text{preserved},l}^{*}$ represents the selected subset of shots to be preserved for that layer. where $\text{TopK}$ selects the subset of shots with the highest importance scores such that the total token count fits within the budget defined by $N_{\text{budget}} \approx r_p \times |KV_{\text{prefill}}|$. The second equation enforces the memory constraint: the total number of tokens ($k_i$) in the selected shots must not exceed the allocated budget, which is determined by the prefill compression ratio $r_p$ multiplied by the original KV cache size.

For the decoding phase, addressing **Observation 6** (Long-Context Generation Sensitivity), we employ a dynamic token-level compression to manage the evolving context independently for each layer:

$$\text{Score}_{\text{decoding}}^{l}(t) = \sum_{h=1}^{H} \alpha_{t,h}^{l} \quad (7)$$

Here, for a previously generated token $t$, $\alpha_{t,h}^{l}$ is similarly defined as the attention weight assigned by the query vector of the current token being generated to the key vector of token $t$, within head $h$ at layer $l$. Thus, $\text{Score}_{\text{decoding}}^{l}(t)$ represents the total attention received by token $t$ at layer $l$ from the current generation step.

Given a decoding compression ratio $r_d$, we select the tokens with the highest scores to preserve. The compressed decoding KV cache $KV_{\text{decoding},l}^{C}$ retains only the top-$k$ tokens where $k = r_d \times |KV_{\text{decoding}}|$, effectively maintaining

the most influential context tokens while reducing memory usage:

$$KV_{\text{decoding},l}^{C} = \text{TopK}(KV_{\text{decoding}}, \text{Score}_{\text{decoding}}^{l}, \\ k = r_d \times |KV_{\text{decoding}}|) \quad (8)$$

Finally, we combine compressed prefill and decoding KV caches to form the total compressed KV cache for each layer:

$$KV_{\text{total},l} = KV_{\text{prefill},l}^{C} \cup KV_{\text{decoding},l}^{C} \quad (9)$$

### 5.2. Empirical Results

In this section, we evaluate the effectiveness of ShotKV on the KVFundaBench. As illustrated in Figure 5, ShotKV consistently achieves top-tier performance across various tasks, demonstrating its robustness as a compression strategy. To provide a more granular analysis, we focus on two particularly challenging scenarios identified in our benchmark analysis: many-shot *Arithmetic Reasoning*, where preserving reasoning chains is critical, and *Long-Context Generation*, which demands sustained coherence over extended outputs. We additionally report a non-ICL generalization study on HotpotQA.

**Baseline.** For *LG-GSM8K* evaluation, we employ three state-of-the-art unified compression methods as baselines: StreamingLLM (Xiao et al., 2024), H2O (Zhang et al., 2023b), and PyramidInfer (Yang et al., 2024). We conduct experiments using LLaMA-3-8B-Instruct (Dubey et al., 2024) on the *LG-GSM8K* benchmark (Liu et al., 2024d), maintaining consistent parameters with Observation 6 ($K = 35, T = 20$). For many-shot *Arithmetic Reasoning* experiments, we follow the configuration detailed in Observation 4.

**Main Results.** As shown in Table 2, ShotKV achieves 47.33% accuracy on *LG-GSM8K* at 40% compression, surpassing the FullKV baseline. Furthermore, on many-shot *Arithmetic Reasoning* (Table 4), ShotKV maintains **80.37%** accuracy even at a 10% ratio. **This significantly outperforms a *Random Shot* baseline (51.34%), where the >29% gap confirms the efficacy of our attention-based semantic unit preservation.**

This superior performance can be attributed to two key design choices: (1) the preservation of complete shot examples ensures the semantic coherence necessary for mathematical reasoning, and (2) the separation of prefill and decoding phases allows for flexible, task-appropriate token retention. These results suggest that our shot-aware strategy is particularly effective for tasks requiring complex reasoning chains. Appendix D.3 provides more detailed ablation analysis on *LG-GSM8K*.

**Latency and Throughput** We further compare the inference efficiency of ShotKV and the FullKV baseline in terms of latency and throughput under different input and output

*Table 3.* LLaMA-3-8B-Instruct on HotpotQA under 10% and 30% compression. ShotKV adapts to this non-ICL document-QA setting by treating each sentence as a semantic unit.

| Comp. % | FullKV | StreamingLLM | H2O | SnapKV | PyramidKV | ChunkKV | ShotKV (Ours) |
|---------|--------|--------------|-----|--------|-----------|---------|---------------|
| 10% | 45.55 | 40.27 | 40.84 | 43.36 | 43.80 | 43.27 | 43.60 |
| 30% | | 42.18 | 42.75 | 44.12 | 44.56 | 44.05 | 44.38 |

*Table 4.* KV cache compression methods' performance on Many-shot *Arithmetic Reasoning*

| Method & Comp. % | 100% | 40% | 30% | 20% | 10% |
|------------------|------|-----|-----|-----|-----|
| FullKV | 82.35 | - | - | - | - |
| StreamingLLM | - | 80.37 | 78.35 | 75.37 | 74.32 |
| H2O | - | 78.32 | 79.32 | 74.28 | 51.27 |
| PyramidKV | - | 78.34 | 79.34 | 78.32 | 70.37 |
| SnapKV | - | 79.35 | 80.38 | 79.34 | 68.27 |
| ChunkKV | - | 78.32 | 79.32 | 78.35 | 79.32 |
| Random Shot | - | 72.50 | 68.23 | 62.50 | 51.34 |
| ShotKV(Ours) | - | **81.07** | **80.82** | **80.57** | **80.37** |

*Table 5.* Latency and throughput comparison among ShotKV, FullKV, and representative compression baselines under different input-output configurations. Tested on the A40 server with batch size 1. Percentages show the relative gain over FullKV.

| Method | Sequence Length | | Performance Metrics | |
|--------|-------|--------|-----------|----------------|
| | Input | Output | Latency(s) ↓ | Throughput(T/S) ↑ |
| FullKV | 4096 | 4096 | 175.50 | 37.73 |
| ChunkKV | 4096 | 4096 | 160.32 (8.6%) | 41.30 (9.5%) |
| SnapKV | 4096 | 4096 | 163.45 (6.9%) | 40.51 (7.4%) |
| ShotKV | 4096 | 4096 | 162.85 (7.2%) | 41.12 (9.0%) |
| FullKV | 8192 | 4096 | 183.42 | 55.93 |
| ShotKV | 8192 | 4096 | 162.78 (11.3%) | 63.24 (13.1%) |

sequence lengths. These results demonstrate that ShotKV not only maintains model performance under aggressive KV cache compression, but also brings tangible efficiency benefits for long-context inference.

**Generalization to Non-ICL Tasks (HotpotQA).** For a document QA setting without few-shot ICL, we adapt ShotKV by treating each sentence as a coherent semantic unit (analogous to a *shot*). Even under an aggressive 10% compression ratio on LLaMA-3-8B-Instruct, ShotKV remains competitive with the best-performing method, as shown in Table 3.

## 6. Limitations

**Deployment scope.** Our method requires direct access to the KV cache, so it targets self-hosted or open-weight deployments (e.g., LLaMA, Mistral, DeepSeek, Qwen) rather than closed API-only systems. This setting matches all KV-compression baselines we compare against and is the regime where KV-level intervention is actionable.

**Scoring is a practical heuristic.** The attention-derived score in ShotKV is an effective operational signal for ranking semantic units under a budget; it is not a principled semantic-importance oracle.

**Orthogonal compression axes.** Our scope is token-level retention; complementary axes such as KV quantization (Li et al., 2025) and broader capability-preservation analyses (Tang et al., 2025) can be combined with ShotKV but are out of scope here.

**Prompt-structure dependence.** ShotKV is most effective when the prompt contains recoverable semantic units (few-shot exemplars or sentence-level chunks). The non-ICL HotpotQA experiment (Table 3) shows the design transfers to document QA via sentence-as-shot adaptation, but extension to fully zero-shot, unstructured long-document settings (e.g., book-scale summarization) remains open. An extended discussion of these limitations is provided in Section B.

## 7. Conclusion

This paper presents KVFundaBench, a benchmark for systematically evaluating the effects of KV cache compression on various fundamental LLM capabilities. Our findings reveal that performance degradation is highly task dependent, with **High-Density Reasoning** (e.g., arithmetic reasoning) and long-context generation being particularly sensitive (*Task-Dependent Degradation* and *Long-Context Generation Sensitivity*). We also highlight that compression sensitivity is influenced by a confluence of factors, including inherent model characteristics such as training dynamics—**where we identify the unique robustness of reasoning models like DeepSeek-R1** (*Model-Type Robustness*)—prompt-level attributes like length (*Prompt Length Vulnerability*), and the reliance on in-context examples (*Prompt-Gain Sensitivity*). Crucially, we demonstrate the importance of preserving the semantic integrity of prompt components, especially at a chunk or shot level, for complex tasks where current token-level methods often struggle (*Chunk-Level Superiority*).

Based on these insights, we developed ShotKV, an insight-driven strategy designed to validate our findings regarding the necessity of semantic preservation. By distinctively managing prefill and decoding phases and prioritizing shot-level **Semantic Units**, ShotKV effectively mitigates information loss in sensitive tasks. The empirical success of ShotKV—**achieving significant accuracy recovery while delivering tangible latency reductions**—confirms the critical role of semantic integrity in reasoning tasks and underscores the potential for more nuanced, task-aware compression strategies.

## Impact Statement

This work advances the field of efficient large language model deployment through systematic analysis and improvement of KV cache compression techniques. Our research has several potential societal impacts:

First, by enabling more efficient memory usage in LLMs while maintaining performance, our work contributes to reducing the computational resources and energy consumption required for AI deployment. This has positive environmental implications and makes AI technology more accessible to researchers and organizations with limited computing resources.

Second, our proposed ShotKV method specifically improves performance on long-context arithmetic reasoning tasks, which could enhance the practical applications of LLMs in education, scientific computing, and other fields requiring complex mathematical reasoning. This could lead to more reliable AI-assisted learning and problem-solving tools.

However, we acknowledge that making LLMs more efficient could accelerate their widespread adoption, potentially raising concerns about AI's impact on employment and privacy. While our work focuses on technical improvements, we encourage the research community to carefully consider these broader implications when deploying such technologies.

We believe the benefits of more efficient and capable AI systems outweigh potential risks, particularly as our work promotes more sustainable and accessible AI development. Nevertheless, we emphasize the importance of responsible deployment and continued ethical consideration in the application of these technologies.

## Acknowledgments

This work was partially supported by the National Natural Science Foundation of China under Grant No. 62272122, and Hong Kong CRF grants under Grant No. C7004-22G and C6015-23G. This work was also supported by the National Natural Science Foundation of China (Grant No. 62506318); Guangdong Provincial Department of Education Project (Grant No. 2024KQNCX028); CAAI-Ant Group Research Fund; Scientific Research Projects for the Higher-educational Institutions (Grant No. 2024312096), Education Bureau of Guangzhou Municipality; and Guangzhou-HKUST(GZ) Joint Funding Program (Grant No. 2025A03J3957), Education Bureau of Guangzhou Municipality.

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

## A. Code Availability

The implementation of ShotKV, KVFundaBench evaluation scripts, and the configuration files used in our experiments are released at `https://github.com/Zefan-Cai/KVCache-Factory`. KVFundaBench extends this open-source KV cache compression framework with additional fundamental-capability benchmarks (world knowledge, commonsense reasoning, arithmetic reasoning, code generation, safety, and long-context generation) and the integrated ShotKV compression strategy. All experiments use publicly available models (e.g., LLaMA-3.1, Mistral-7B, DeepSeek-R1, Qwen3) and standard academic datasets evaluated through the `lm-evaluation-harness` and `KVpress` interfaces. Detailed hyperparameters and experimental configurations are provided in Appendix D.

## B. Extended Limitations

While KVFundaBench provides a comprehensive evaluation of KV cache compression on fundamental capabilities and ShotKV demonstrates significant improvements, there are a few aspects that warrant further exploration. First, our analysis primarily focuses on open-weights models (e.g., LLaMA-3, Mistral, DeepSeek) where accessing and manipulating the KV cache is straightforward. Extending these findings to proprietary, API-based models remains a direction for future research due to current access restrictions on internal inference states. Second, although ShotKV achieves superior performance by preserving semantic units, it relies on a heuristic for shot importance that, while effective, introduces a negligible calculation step during the prefill phase. As shown in our efficiency analysis, the throughput gains from compression significantly outweigh this cost, though further kernel-level optimizations could be explored. Finally, while we cover five distinct categories of fundamental abilities, the landscape of LLM capabilities is vast. Future work could extend our framework to assess more specialized domains or multimodal tasks as efficient inference techniques evolve.

## C. Extended Related Work

**Key–value Cache Optimization Techniques** KV cache is the core component in LLM inference, which avoids repetitive computations by caching Key and Value vectors. However, the cost of caching KV increases exponentially with the expansion of the model size and the length of the context (Pope et al., 2023). Some approaches have been published to alleviate the problem. For example, KV Compression designs efficient content selection strategies to filter and manage tokens (Zhang et al., 2023b; Adnan et al., 2024). Some methods identify important tokens by focusing on high attention allocation (Li et al., 2024b), while others optimize token selection by combining attention scores with value vector norms to improve importance evaluation (Guo et al., 2024). Techniques like PyramidInfer reduce critical contexts layer by layer based on the distribution of attention scores (Yang et al., 2024), and StreamingLLM preserves attention sinks to maintain stable performance in extended sequences (Xiao et al., 2024). Researchers reduce storage costs by merging similar context representations and solving input disturbances caused by compression (Zhang et al., 2024b). For example, CaM (Zhang et al., 2024b) works by integrating the KV cache to be dropped into the retained cache in proportion to the attention weight. In addition, Yao et al. (2025) proposes CacheBlend to achieve a selective KV recompute. Only partial KVs of crucial tokens are updated to reduce the delay in the prefill stage and increase the throughput. In addition, the dynamic budget allocation method is also used to optimize the KV cache, which adjusts the resource allocation in real time according to the importance of the context, providing a balance between performance and efficiency in multitask inference scenarios (Cai et al., 2024; Feng et al., 2025; Kim et al., 2025; Devoto et al., 2025; Liu et al., 2025a; Zhu et al., 2025).Wu et al. (2025) proposes a prefill-decoding separation strategy to optimize the KV cache compression.

**Evaluation of LLMs' Fundamental Abilities** Accurately evaluating the fundamental capabilities of large language models is crucial to understand their true potential and limitations. The evaluation typically spans across several key dimensions: world knowledge tasks like MMLU (Hendrycks et al., 2021a),BBH (Suzgun et al., 2023) assess models' grasp of diverse domains through multiple-choice questions; commonsense reasoning tasks such as CSQA (Talmor et al., 2019) evaluate inference and context understanding abilities; arithmetic reasoning benchmarks like GSM8K (Cobbe et al., 2021) test mathematical problem-solving capabilities through step-by-step reasoning; code generation tasks including HumanEval (Chen et al., 2021; Yu et al., 2026) measure the ability to generate functionally correct code; and safety evaluations using benchmarks like JailBreakV (Luo et al., 2024) assess models' robustness against harmful content generation. Additionally, long-context benchmarks such as LongBench (Bai et al., 2024; 2025) and Need-In-A-Haystack (NIAH) (Kamradt, 2023) aiming to evaluate models' long-context summarization and retrieval capabilities. Furthermore, LongGenBench (Liu et al., 2024d) evaluates the models' ability to process and generate responses for extended input

sequences. And recently, in-context many-shot learning has been recognized as a long-context reasoning paradigm (Agarwal et al., 2024; Tang et al., 2026), which considers the number of shots as a critical factor in the performance of LLM. Although these tasks typically employ automatic evaluation metrics for standardization, KV cache compression may introduce unique challenges, particularly in tasks requiring complex reasoning chains or extensive knowledge retrieval.

**KV cache sharing** Recent work has explored various strategies for sharing KV caches across transformer layers. The Layer Condensed KV Cache (LCKV) (Wu & Tu, 2024) computes the KV only for the top layer and pairs them with queries from all layers, while optionally retaining standard attention for a few top and bottom layers to mitigate performance degradation. Similarly, You Only Cache Once (YOCO) (Sun et al., 2024) computes KVs exclusively for the top layer but pairs them with queries from only the top half of layers, employing efficient attention in the bottom layers to maintain a constant cache size. In contrast, Cross-Layer Attention (CLA) (Brandon et al., 2024) divides layers into groups, pairing queries from all layers in each group with KVs from that group's bottom layer. MiniCache (Liu et al., 2024b) introduces a novel method that merges KV caches in layering while enabling recovery during compute-in-place operations, optimizing the size of the KV cache. These methods illustrate various trade-offs between computation, memory usage, and model performance when sharing KV caches across transformer layers.

**Prompting Compression** Recent advances in prompt compression have yielded innovative approaches to information density optimization in natural language processing. Research by Wingate et al. (2022) demonstrates how soft prompting techniques can achieve higher information density per token. Building upon this foundation, AutoCompressor (Chevalier et al., 2023) leverages soft prompts to both condense input sequences and expand model context windows. Parallel developments by Zhou et al. (2023) and Wang et al. (2023) showcase iterative summarization strategies using LLMs, establishing persistent memory mechanisms particularly beneficial for narrative construction and conversational systems. The progressive development of the LLMLingua framework (Jiang et al., 2023b; 2024; Fei et al., 2024) has advanced prompt compression capabilities across extended context processing, logical reasoning, and retrieval-augmented generation. Notable contributions from Fei et al. (2024) demonstrate effective context management through automated segmentation and semantic condensation using pre-trained language models.

**General Tasks** General tasks refer to evaluating the overall performance of LLMs under mathematical inference, logic reasoning, and common knowledge. GSM8K (Cobbe et al., 2021) and MMLU (Hendrycks et al., 2021a) are representative tasks. The former focuses on the step-by-step reasoning ability of mathematical problem solving, while the latter covers assessment of common sense and expertise in multiple areas. Besides, MATH (Hendrycks et al., 2021b) spans various mathematical fields, ranging from elementary algebra to calculus, aiming to improve the mathematical problem-solving capabilities of LLMs. Meanwhile, MathQA (Amini et al., 2019) is a large-scale dataset comprising approximately 37,000 multiple-choice questions with precise annotations, designed to enhance the interpretability and performance of LLMs. In addition, BBH (Suzgun et al., 2023), a subset of BIG-Bench (Srivastava et al., 2023), focuses on challenging tasks. BBH includes multi-step reasoning problems, highlighting the importance of Chain-of-Thought prompting in LLMs. Similarly, CSQA (Talmor et al., 2019) is a task that combines knowledge graph-based multi-step reasoning with conversational capabilities. CSQA emphasizes inference and context understanding grounded in knowledge graphs. Normally, the general tasks apply automatic evaluation metrics (e.g. multi-choice accuracy) to ensure comparability and standardization. However, optimization strategies like KV cache compression may introduce challenges in executing the mentioned tasks. Filtering and dropping of contexts are involved in the compression strategy which may lead to an intermediate inference steps missing. In addition, in tasks such as MMLU that are highly dependent on knowledge coverage, compression may weaken the model's ability to capture long context or rare domain knowledge (Yuan et al., 2024).

**Security Tasks** Security tasks focus on assessing the robustness and protections of LLMs against harmful content, including truthfulness (Lin et al., 2022), toxicity (Hartvigsen et al., 2022), and bias (Liang et al., 2021; Tang et al., 2026). Recently, researchers noticed the weakness of LLMs in adversarial prompts (Zhu et al., 2024; TANG et al., 2026), especially in generating illegal or inappropriate content under jailbreak prompts. Shen et al. (2024) analyze the jailbreak prompts in real cases to reveal the failure of model security mechanism under complex malicious input. Meanwhile, Deng et al. (2024) demonstrates the multilingual jailbreak makes model security in low-resource languages easier to bypass, significantly increasing the probability that users of low-resource languages will generate insecure content. Similar to general tasks, KV optimization techniques can cause the model to ignore potential security threats when dealing with jailbreak prompts, thereby improving the success rate of adversarial prompts (Li et al., 2024a).

**Code Generation Tasks**   Code generation tasks test the capacities of LLMs to generate code, which not only requires that the model can generate syntactic code based on natural language description but also has certain logical reasoning abilities. HumanEval (Chen et al., 2021) and MBPP (Austin et al., 2021) are the commonly used benchmarks. They measure the functional correctness of the model by testing the results of the code's execution.

**Long-context Tasks**   Recent developments in evaluating long-context models have produced a comprehensive ecosystem of benchmarks, focusing on both comprehension depth and retrieval efficiency. In the comprehension domain, ∞-Bench (Zhang et al., 2024a) has established new standards by crafting evaluation scenarios exceeding 100,000 tokens, while LongBench (Bai et al., 2024; 2025) introduced multilingual assessment frameworks spanning document comprehension, text synthesis, and programming tasks. Further enriching this landscape, ZeroSCROLLS (Shaham et al., 2023) and L-Eval (An et al., 2024) have expanded evaluation criteria to encompass real-world applications, particularly in query-based content summarization. The emergence of many-shot learning as a distinct paradigm for extended context processing (Agarwal et al., 2024) has added another dimension to this field. Notable contributions from LongGenBench (Liu et al., 2024d) have advanced evaluation methodologies by combining extensive response generation requirements with efficient, cost-effective quality metrics.

The development of retrieval-focused benchmarks has taken a distinct approach, predominantly utilizing constructed datasets that enable precise experimental control, particularly in managing input sequence lengths. This methodology helps neutralize variations in model performance stemming from differences in training approaches. Substantial research efforts have yielded specialized synthetic frameworks for assessing retrieval capabilities (Kamradt, 2023; Mohtashami & Jaggi, 2023; Li et al., 2023; Liu et al., 2024c; Hsieh et al., 2024), while concurrent investigations have revealed the broader implications of extended context processing for enhanced reasoning capabilities (Tay et al., 2021).

# D. Experiment Details

## D.1. Detail Results

This section provide the detailed results of experiments in this paper, the results are shown in the format of $x_y$, where $x$ is the performance of the method and $y$ is the $\Delta P$ from the Equation (1).

**Observation 1.   KV cache compression methods show task-dependent performance degradation, WK and CSR are more robust to KV cache compression.**

The detailed results of different KV cache compression methods are shown in Table 7, different tasks exhibit notably varied sensitivities to KV cache compression, particularly under aggressive compression ratios. At a 10% compression ratio, MMLU demonstrates remarkable resilience with less than 1% average performance degradation, while GSM8K experiences a severe average performance drop exceeding 35%. Other tasks show moderate to significant degradation, ranging from 6.5% to 17.2%. This substantial variation in compression sensitivity across tasks suggests that the effectiveness of KV cache compression is highly task-dependent, necessitating careful consideration of the specific task requirements when determining appropriate compression ratios.

The Table 6 compares the performance of R1-Distill-Llama-8B and LLaMA-3.1-8B-Instruct under different compression ratios. R1-Distill-Llama-8B demonstrates more robust performance under compression compared to LLaMA-3.1-8B-Instruct. While both models start with similar baseline performance (0.6938 vs 0.7945), R1-Distill shows significantly less performance degradation under aggressive compression. Specifically, at 30% compression ratio, R1-Distill maintains a performance of 0.6407 (-7.66%), while LLaMA-3.1-8B-Instruct drops to 0.7469 (-6.00%). The difference becomes more pronounced at 10% compression ratio, where R1-Distill achieves 0.5840 (-15.82%) compared to LLaMA-3.1-8B-Instruct's sharp decline to 0.5143 (-35.30%). This suggests that the multi-step reasoning capabilities of R1-Distill contribute to its resilience against aggressive KV cache compression, particularly in maintaining reasoning coherence under limited context conditions.

On safety-focused evaluations, we observe that aggressive compression can disproportionately degrade performance, plausibly because compression may discard or fragment subtle safety-critical keywords and phrases present in system prompts; this disruption can weaken safety constraints during generation.

*Table 6.* Performance Comparison of Different KV Cache Compression Methods on Instruction-Tuning Model and Multi-Step Reasoning Model

| Benchmark | Ratio | StreamingLLM | H2O | SnapKV | PyramidKV | ChunkKV | Average ↑ |
|---|---|---|---|---|---|---|---|
| | Baseline | | | R1-Distill-Llama-8B FullKV: 0.6938 | | | |
| | 90% | $0.7167_{(+3.30\%)}$ | $0.6900_{(-0.55\%)}$ | $0.6933_{(-0.07\%)}$ | $0.7100_{(+2.34\%)}$ | $0.6867_{(-1.02\%)}$ | $0.6993_{(+0.79\%)}$ |
| | 80% | $0.6867_{(-1.02\%)}$ | $0.6933_{(-0.07\%)}$ | $0.6933_{(-0.07\%)}$ | $0.7067_{(+1.86\%)}$ | $0.6767_{(-2.47\%)}$ | $0.6913_{(-0.36\%)}$ |
| | 70% | $0.6933_{(-0.07\%)}$ | $0.6633_{(-4.40\%)}$ | $0.7100_{(+2.34\%)}$ | $0.7100_{(+2.34\%)}$ | $0.7000_{(+0.89\%)}$ | $0.6953_{(+0.22\%)}$ |
| *R1-AR* | 60% | $0.6833_{(-1.51\%)}$ | $0.6900_{(-0.55\%)}$ | $0.6900_{(-0.55\%)}$ | $0.7133_{(+2.81\%)}$ | $0.7067_{(+1.86\%)}$ | $0.6967_{(+0.42\%)}$ |
| | 50% | $0.6700_{(-3.43\%)}$ | $0.6967_{(+0.42\%)}$ | $0.7067_{(+1.86\%)}$ | $0.7000_{(+0.89\%)}$ | $0.6867_{(-1.02\%)}$ | $0.6920_{(-0.26\%)}$ |
| | 40% | $0.6767_{(-2.47\%)}$ | $0.6800_{(-1.99\%)}$ | $0.5967_{(-13.99\%)}$ | $0.6967_{(+0.42\%)}$ | $0.7133_{(+2.81\%)}$ | $0.6727_{(-3.04\%)}$ |
| | 30% | $0.6600_{(-4.87\%)}$ | $0.5900_{(-14.96\%)}$ | $0.5833_{(-15.93\%)}$ | $0.6700_{(-3.43\%)}$ | $0.7000_{(+0.89\%)}$ | $0.6407_{(-7.66\%)}$ |
| | 20% | $0.6200_{(-10.64\%)}$ | $0.4933_{(-28.90\%)}$ | $0.5633_{(-18.81\%)}$ | $0.6833_{(-1.51\%)}$ | $0.6533_{(-5.84\%)}$ | $0.6026_{(-13.14\%)}$ |
| | 10% | $0.5167_{(-25.53\%)}$ | $0.5567_{(-19.76\%)}$ | $0.5767_{(-16.88\%)}$ | $0.6267_{(-9.67\%)}$ | $0.6433_{(-7.28\%)}$ | $0.5840_{(-15.82\%)}$ |
| | Baseline | | | LLaMA-3.1-8B-Instruct FullKV: 0.7945 | | | |
| | 90% | $0.7695_{(-3.10\%)}$ | $0.7923_{(-0.30\%)}$ | $0.7839_{(-1.30\%)}$ | $0.7854_{(-1.10\%)}$ | $0.7824_{(-1.50\%)}$ | $0.7827_{(-1.50\%)}$ |
| | 80% | $0.7642_{(-3.80\%)}$ | $0.7938_{(-0.10\%)}$ | $0.7824_{(-1.50\%)}$ | $0.7900_{(-0.60\%)}$ | $0.7824_{(-1.50\%)}$ | $0.7826_{(-1.50\%)}$ |
| | 70% | $0.7642_{(-3.80\%)}$ | $0.7900_{(-0.60\%)}$ | $0.7923_{(-0.30\%)}$ | $0.7983_{(+0.50\%)}$ | $0.7809_{(-1.70\%)}$ | $0.7851_{(-1.20\%)}$ |
| *AR* | 60% | $0.7650_{(-3.70\%)}$ | $0.7809_{(-1.70\%)}$ | $0.7885_{(-0.80\%)}$ | $0.7923_{(-0.30\%)}$ | $0.7885_{(-0.80\%)}$ | $0.7830_{(-1.50\%)}$ |
| | 50% | $0.7657_{(-3.60\%)}$ | $0.7854_{(-1.10\%)}$ | $0.7847_{(-1.20\%)}$ | $0.7854_{(-1.10\%)}$ | $0.7824_{(-1.50\%)}$ | $0.7807_{(-1.70\%)}$ |
| | 40% | $0.7491_{(-5.70\%)}$ | $0.7688_{(-3.20\%)}$ | $0.7756_{(-2.40\%)}$ | $0.7839_{(-1.30\%)}$ | $0.7763_{(-2.30\%)}$ | $0.7707_{(-3.00\%)}$ |
| | 30% | $0.7051_{(-11.20\%)}$ | $0.7225_{(-9.10\%)}$ | $0.7619_{(-4.10\%)}$ | $0.7718_{(-2.90\%)}$ | $0.7733_{(-2.70\%)}$ | $0.7469_{(-6.00\%)}$ |
| | 20% | $0.6384_{(-19.70\%)}$ | $0.6406_{(-19.40\%)}$ | $0.6884_{(-13.40\%)}$ | $0.7142_{(-10.10\%)}$ | $0.7763_{(-2.30\%)}$ | $0.6916_{(-13.00\%)}$ |
| | 10% | $0.4784_{(-39.80\%)}$ | $0.4503_{(-43.30\%)}$ | $0.5034_{(-36.60\%)}$ | $0.4829_{(-39.20\%)}$ | $0.6566_{(-17.40\%)}$ | $0.5143_{(-35.30\%)}$ |

**Observation 2. Multi-step reasoning LLMs are more robust to KV cache compression.** As shown in Table 8, while instruct-tuned models achieve superior baseline performance (0.7945 vs 0.5122), they demonstrate heightened sensitivity to KV cache compression. This sensitivity becomes particularly pronounced at aggressive compression ratios. At 10% compression ratio, instruct-tuned models suffer an average performance degradation of 35.3% (from 0.7945 to 0.5143), nearly double the degradation observed in non-instruct-tuned models which show a 17.2% drop (from 0.5122 to 0.4244). In contrast, R1-Distill-Llama-8B shows better resilience to compression, with only a 15.82% performance drop (from 0.6938 to 0.5840) at 10% compression ratio. This pattern suggests that while instruction tuning enhances model capabilities, it also makes the model more dependent on maintaining complete context information. However, models trained with multi-step reasoning capabilities like R1-Distill demonstrate better robustness against aggressive compression, likely due to their enhanced ability to maintain reasoning coherence even with limited context. We hypothesize that the reinforcement learning objective that explicitly incentivizes multi-step reasoning in DeepSeek-R1 yields more structured and robust internal representations of reasoning chains, making them less fragile to KV cache compression.

**Observation 3. Short prompt length is more sensitive to KV cache compression.** As demonstrated in Table 9, the impact of KV cache compression varies significantly with the number of shots in the prompt. One-shot prompts show extreme vulnerability to aggressive compression, with performance plummeting from 0.7149 to 0.0452 (a 93.7% drop) at 10% compression ratio. This sensitivity gradually decreases as the number of shots increases. For instance, at the same compression ratio, 4-shot prompts show a 46.2% performance drop (from 0.7597 to 0.4088), while 8-shot prompts demonstrate relatively better resilience with a 35.3% reduction (from 0.7945 to 0.5143). This pattern suggests that longer prompts with more examples provide redundancy that helps maintain model performance under compression, while shorter prompts lack this buffer against information loss.

**Observation 4. Chunk-level compression is more effective for long-context structured reasoning tasks.** As shown in Table 10, ChunkKV demonstrates superior robustness across different compression ratios, particularly under aggressive compression settings. While other methods show significant performance degradation at 10% compression ratio (StreamingLLM: -9.8%, H2O: -37.8%, SnapKV: -17.1%, PyramidKV: -14.6%), ChunkKV maintains relatively stable performance with only a -3.7% drop. This stark contrast in performance suggests that chunk-level compression better preserves the essential contextual information needed for complex reasoning tasks. The method's effectiveness likely stems from its ability to maintain the structural integrity of related context segments, which is particularly crucial for tasks requiring extended logical reasoning and arithmetic operations.

*Table 7.* Performance Comparison of Different KV Cache Compression Methods on KVFundaBench

| Benchmark | Ratio | StreamingLLM | H2O | SnapKV | PyramidKV | ChunkKV | Average ↑ |
|---|---|---|---|---|---|---|---|
| | Baseline | | | FullKV: 0.6882 | | | |
| WK | 90% | $0.6882_{(+0.00\%)}$ | $0.6882_{(+0.00\%)}$ | $0.6882_{(+0.00\%)}$ | $0.6882_{(+0.00\%)}$ | $0.6882_{(+0.00\%)}$ | $0.6882_{(+0.00\%)}$ |
| | 80% | $0.6882_{(+0.00\%)}$ | $0.6882_{(+0.00\%)}$ | $0.6882_{(+0.00\%)}$ | $0.6882_{(+0.00\%)}$ | $0.6882_{(+0.00\%)}$ | $0.6882_{(+0.00\%)}$ |
| | 70% | $0.6881_{(-0.01\%)}$ | $0.6882_{(+0.00\%)}$ | $0.6882_{(+0.00\%)}$ | $0.6882_{(+0.00\%)}$ | $0.6882_{(+0.00\%)}$ | $0.6882_{(+0.00\%)}$ |
| | 60% | $0.6881_{(-0.01\%)}$ | $0.6882_{(+0.00\%)}$ | $0.6882_{(+0.00\%)}$ | $0.6882_{(+0.00\%)}$ | $0.6882_{(+0.00\%)}$ | $0.6882_{(+0.00\%)}$ |
| | 50% | $0.6881_{(-0.01\%)}$ | $0.6882_{(+0.00\%)}$ | $0.6882_{(+0.00\%)}$ | $0.6882_{(+0.00\%)}$ | $0.6882_{(+0.00\%)}$ | $0.6882_{(+0.00\%)}$ |
| | 40% | $0.6879_{(-0.04\%)}$ | $0.6882_{(+0.00\%)}$ | $0.6882_{(+0.00\%)}$ | $0.6882_{(+0.00\%)}$ | $0.6882_{(+0.00\%)}$ | $0.6881_{(-0.01\%)}$ |
| | 30% | $0.6876_{(-0.09\%)}$ | $0.6880_{(-0.03\%)}$ | $0.6880_{(-0.03\%)}$ | $0.6882_{(+0.00\%)}$ | $0.6882_{(+0.00\%)}$ | $0.6880_{(-0.03\%)}$ |
| | 20% | $0.6859_{(-0.33\%)}$ | $0.6878_{(-0.06\%)}$ | $0.6880_{(-0.03\%)}$ | $0.6882_{(+0.00\%)}$ | $0.6882_{(+0.00\%)}$ | $0.6876_{(-0.08\%)}$ |
| | 10% | $0.6787_{(-1.38\%)}$ | $0.6852_{(-0.44\%)}$ | $0.6831_{(-0.74\%)}$ | $0.6882_{(0.00\%)}$ | $0.6842_{(-0.58\%)}$ | $0.6839_{(-0.63\%)}$ |
| | Baseline | | | FullKV: 0.7945 | | | |
| AR | 90% | $0.7695_{(-3.10\%)}$ | $0.7923_{(-0.30\%)}$ | $0.7839_{(-1.30\%)}$ | $0.7854_{(-1.10\%)}$ | $0.7824_{(-1.50\%)}$ | $0.7827_{(-1.50\%)}$ |
| | 80% | $0.7642_{(-3.80\%)}$ | $0.7938_{(-0.10\%)}$ | $0.7824_{(-1.50\%)}$ | $0.7900_{(-0.60\%)}$ | $0.7824_{(-1.50\%)}$ | $0.7826_{(-1.50\%)}$ |
| | 70% | $0.7642_{(-3.80\%)}$ | $0.7900_{(-0.60\%)}$ | $0.7923_{(-0.30\%)}$ | $0.7983_{(+0.50\%)}$ | $0.7809_{(-1.70\%)}$ | $0.7851_{(-1.20\%)}$ |
| | 60% | $0.7650_{(-3.70\%)}$ | $0.7809_{(-1.70\%)}$ | $0.7885_{(-0.80\%)}$ | $0.7923_{(-0.30\%)}$ | $0.7885_{(-0.80\%)}$ | $0.7830_{(-1.50\%)}$ |
| | 50% | $0.7657_{(-3.60\%)}$ | $0.7854_{(-1.10\%)}$ | $0.7847_{(-1.20\%)}$ | $0.7854_{(-1.10\%)}$ | $0.7824_{(-1.50\%)}$ | $0.7807_{(-1.70\%)}$ |
| | 40% | $0.7491_{(-5.70\%)}$ | $0.7688_{(-3.20\%)}$ | $0.7756_{(-2.40\%)}$ | $0.7839_{(-1.30\%)}$ | $0.7763_{(-2.30\%)}$ | $0.7707_{(-3.00\%)}$ |
| | 30% | $0.7051_{(-11.20\%)}$ | $0.7225_{(-9.10\%)}$ | $0.7619_{(-4.10\%)}$ | $0.7718_{(-2.90\%)}$ | $0.7733_{(-2.70\%)}$ | $0.7469_{(-6.00\%)}$ |
| | 20% | $0.6384_{(-19.70\%)}$ | $0.6406_{(-19.40\%)}$ | $0.6884_{(-13.40\%)}$ | $0.7142_{(-10.10\%)}$ | $0.7763_{(-2.30\%)}$ | $0.6916_{(-13.00\%)}$ |
| | 10% | $0.4784_{(-39.80\%)}$ | $0.4503_{(-43.30\%)}$ | $0.5034_{(-36.60\%)}$ | $0.4829_{(-39.20\%)}$ | $0.6566_{(-17.40\%)}$ | $0.5143_{(-35.30\%)}$ |
| | Baseline | | | FullKV: 0.7748 | | | |
| CSR | 90% | $0.7748_{(+0.00\%)}$ | $0.7748_{(+0.00\%)}$ | $0.7748_{(+0.00\%)}$ | $0.7748_{(+0.00\%)}$ | $0.7748_{(+0.00\%)}$ | $0.7748_{(+0.00\%)}$ |
| | 80% | $0.7748_{(+0.00\%)}$ | $0.7748_{(+0.00\%)}$ | $0.7748_{(+0.00\%)}$ | $0.7748_{(+0.00\%)}$ | $0.7748_{(+0.00\%)}$ | $0.7748_{(+0.00\%)}$ |
| | 70% | $0.7748_{(+0.00\%)}$ | $0.7748_{(+0.00\%)}$ | $0.7748_{(+0.00\%)}$ | $0.7748_{(+0.00\%)}$ | $0.7748_{(+0.00\%)}$ | $0.7748_{(+0.00\%)}$ |
| | 60% | $0.7748_{(+0.00\%)}$ | $0.7748_{(+0.00\%)}$ | $0.7748_{(+0.00\%)}$ | $0.7748_{(+0.00\%)}$ | $0.7748_{(+0.00\%)}$ | $0.7748_{(+0.00\%)}$ |
| | 50% | $0.7748_{(+0.00\%)}$ | $0.7748_{(+0.00\%)}$ | $0.7748_{(+0.00\%)}$ | $0.7748_{(+0.00\%)}$ | $0.7748_{(+0.00\%)}$ | $0.7748_{(+0.00\%)}$ |
| | 40% | $0.7748_{(+0.00\%)}$ | $0.7748_{(+0.00\%)}$ | $0.7748_{(+0.00\%)}$ | $0.7748_{(+0.00\%)}$ | $0.7748_{(+0.00\%)}$ | $0.7748_{(+0.00\%)}$ |
| | 30% | $0.7748_{(+0.00\%)}$ | $0.7748_{(+0.00\%)}$ | $0.7748_{(+0.00\%)}$ | $0.7748_{(+0.00\%)}$ | $0.7748_{(+0.00\%)}$ | $0.7748_{(+0.00\%)}$ |
| | 20% | $0.7174_{(-7.40\%)}$ | $0.7748_{(+0.00\%)}$ | $0.7740_{(-0.10\%)}$ | $0.7748_{(+0.00\%)}$ | $0.7699_{(-0.60\%)}$ | $0.7622_{(-1.60\%)}$ |
| | 10% | $0.6806_{(-12.20\%)}$ | $0.7510_{(-3.10\%)}$ | $0.7191_{(-7.20\%)}$ | $0.7723_{(-0.30\%)}$ | $0.7002_{(-9.60\%)}$ | $0.7246_{(-6.50\%)}$ |
| | Baseline | | | FullKV: 0.8895 | | | |
| SA | 90% | $0.8893_{(-0.00\%)}$ | $0.8890_{(-0.10\%)}$ | $0.8894_{(-0.00\%)}$ | $0.8893_{(-0.00\%)}$ | $0.8896_{(+0.00\%)}$ | $0.8893_{(-0.00\%)}$ |
| | 80% | $0.8878_{(-0.20\%)}$ | $0.8885_{(-0.10\%)}$ | $0.8895_{(+0.00\%)}$ | $0.8891_{(-0.00\%)}$ | $0.8894_{(-0.00\%)}$ | $0.8889_{(-0.10\%)}$ |
| | 70% | $0.8872_{(-0.30\%)}$ | $0.8879_{(-0.20\%)}$ | $0.8896_{(+0.00\%)}$ | $0.8889_{(-0.10\%)}$ | $0.8895_{(+0.00\%)}$ | $0.8886_{(-0.10\%)}$ |
| | 60% | $0.8845_{(-0.60\%)}$ | $0.8848_{(-0.50\%)}$ | $0.8892_{(-0.00\%)}$ | $0.8887_{(-0.10\%)}$ | $0.8899_{(+0.00\%)}$ | $0.8874_{(-0.20\%)}$ |
| | 50% | $0.8849_{(-0.50\%)}$ | $0.8749_{(-1.60\%)}$ | $0.8886_{(-0.10\%)}$ | $0.8884_{(-0.10\%)}$ | $0.8894_{(-0.00\%)}$ | $0.8852_{(-0.50\%)}$ |
| | 40% | $0.8734_{(-1.80\%)}$ | $0.8557_{(-3.80\%)}$ | $0.8880_{(-0.20\%)}$ | $0.8877_{(-0.20\%)}$ | $0.8900_{(+0.10\%)}$ | $0.8790_{(-1.20\%)}$ |
| | 30% | $0.8329_{(-6.40\%)}$ | $0.8015_{(-9.90\%)}$ | $0.8858_{(-0.40\%)}$ | $0.8899_{(+0.00\%)}$ | $0.8846_{(-0.60\%)}$ | $0.8589_{(-3.50\%)}$ |
| | 20% | $0.6501_{(-26.90\%)}$ | $0.7178_{(-19.30\%)}$ | $0.8806_{(-1.00\%)}$ | $0.8751_{(-1.60\%)}$ | $0.8902_{(+0.10\%)}$ | $0.8028_{(-9.70\%)}$ |
| | 10% | $0.5314_{(-40.30\%)}$ | $0.6544_{(-26.40\%)}$ | $0.8434_{(-5.20\%)}$ | $0.8556_{(-3.80\%)}$ | $0.8799_{(-1.10\%)}$ | $0.7529_{(-15.40\%)}$ |
| | Baseline | | | FullKV: 0.5122 | | | |
| CG | 90% | $0.5061_{(-1.20\%)}$ | $0.5183_{(+1.20\%)}$ | $0.5122_{(+0.00\%)}$ | $0.5122_{(+0.00\%)}$ | $0.5122_{(+0.00\%)}$ | $0.5122_{(+0.00\%)}$ |
| | 80% | $0.5061_{(-1.20\%)}$ | $0.5183_{(+1.20\%)}$ | $0.5183_{(+1.20\%)}$ | $0.5305_{(+3.60\%)}$ | $0.5061_{(-1.20\%)}$ | $0.5159_{(+0.70\%)}$ |
| | 70% | $0.5000_{(-2.40\%)}$ | $0.5244_{(+2.40\%)}$ | $0.5122_{(+0.00\%)}$ | $0.5183_{(+1.20\%)}$ | $0.5122_{(+0.00\%)}$ | $0.5134_{(+0.20\%)}$ |
| | 60% | $0.5061_{(-1.20\%)}$ | $0.5366_{(+4.80\%)}$ | $0.5366_{(+4.80\%)}$ | $0.5305_{(+3.60\%)}$ | $0.5244_{(+2.40\%)}$ | $0.5268_{(+2.90\%)}$ |
| | 50% | $0.4939_{(-3.60\%)}$ | $0.5427_{(+6.00\%)}$ | $0.5061_{(-1.20\%)}$ | $0.4939_{(-3.60\%)}$ | $0.4878_{(-4.80\%)}$ | $0.5049_{(-1.40\%)}$ |
| | 40% | $0.4817_{(-6.00\%)}$ | $0.5427_{(+6.00\%)}$ | $0.5244_{(+2.40\%)}$ | $0.4939_{(-3.60\%)}$ | $0.5000_{(-2.40\%)}$ | $0.5085_{(-0.70\%)}$ |
| | 30% | $0.4817_{(-6.00\%)}$ | $0.5305_{(+3.60\%)}$ | $0.5000_{(-2.40\%)}$ | $0.4939_{(-3.60\%)}$ | $0.4817_{(-6.00\%)}$ | $0.4976_{(-2.90\%)}$ |
| | 20% | $0.4634_{(-9.50\%)}$ | $0.5061_{(-1.20\%)}$ | $0.4939_{(-3.60\%)}$ | $0.4695_{(-8.30\%)}$ | $0.4878_{(-4.80\%)}$ | $0.4841_{(-5.50\%)}$ |
| | 10% | $0.3659_{(-28.60\%)}$ | $0.4634_{(-9.50\%)}$ | $0.4268_{(-16.70\%)}$ | $0.4207_{(-17.90\%)}$ | $0.4451_{(-13.10\%)}$ | $0.4244_{(-17.20\%)}$ |

### D.2. Matched-setting comparison: retrieval vs. high-density reasoning

To directly substantiate the claim in Section 1 that high-density reasoning is more compression-sensitive than retrieval-oriented long-context tasks, we compare relative degradation under *matched* compression methods and ratios. We use

| Setting | Ratio | StreamingLLM | H2O | SnapKV | PyramidKV | ChunkKV | Average ↑ |
|---|---|---|---|---|---|---|---|
| | Baseline | | | FullKV: 0.7945 | | | |
| w/ Instruct Tuning | 90% | $0.7695_{(-3.10\%)}$ | $0.7923_{(-0.30\%)}$ | $0.7839_{(-1.30\%)}$ | $0.7854_{(-1.10\%)}$ | $0.7824_{(-1.50\%)}$ | $0.7827_{(-1.50\%)}$ |
| | 80% | $0.7642_{(-3.80\%)}$ | $0.7938_{(-0.10\%)}$ | $0.7824_{(-1.50\%)}$ | $0.7900_{(-0.60\%)}$ | $0.7824_{(-1.50\%)}$ | $0.7826_{(-1.50\%)}$ |
| | 70% | $0.7642_{(-3.80\%)}$ | $0.7900_{(-0.60\%)}$ | $0.7923_{(-0.30\%)}$ | $0.7983_{(+0.50\%)}$ | $0.7809_{(-1.70\%)}$ | $0.7851_{(-1.20\%)}$ |
| | 60% | $0.7650_{(-3.70\%)}$ | $0.7809_{(-1.70\%)}$ | $0.7885_{(-0.80\%)}$ | $0.7923_{(-0.30\%)}$ | $0.7885_{(-0.80\%)}$ | $0.7830_{(-1.50\%)}$ |
| | 50% | $0.7657_{(-3.60\%)}$ | $0.7854_{(-1.10\%)}$ | $0.7847_{(-1.20\%)}$ | $0.7854_{(-1.10\%)}$ | $0.7824_{(-1.50\%)}$ | $0.7807_{(-1.70\%)}$ |
| | 40% | $0.7491_{(-5.70\%)}$ | $0.7688_{(-3.20\%)}$ | $0.7756_{(-2.40\%)}$ | $0.7839_{(-1.30\%)}$ | $0.7763_{(-2.30\%)}$ | $0.7707_{(-3.00\%)}$ |
| | 30% | $0.7051_{(-11.20\%)}$ | $0.7225_{(-9.10\%)}$ | $0.7619_{(-4.10\%)}$ | $0.7718_{(-2.90\%)}$ | $0.7733_{(-2.70\%)}$ | $0.7469_{(-6.00\%)}$ |
| | 20% | $0.6384_{(-19.70\%)}$ | $0.6406_{(-19.40\%)}$ | $0.6884_{(-13.40\%)}$ | $0.7142_{(-10.10\%)}$ | $0.7763_{(-2.30\%)}$ | $0.6916_{(-13.00\%)}$ |
| | 10% | $0.4784_{(-39.80\%)}$ | $0.4503_{(-43.30\%)}$ | $0.5034_{(-36.60\%)}$ | $0.4829_{(-39.20\%)}$ | $0.6566_{(-17.40\%)}$ | $0.5143_{(-35.30\%)}$ |
| | Baseline | | | R1-Distill-Llama-8B FullKV: 0.6938 | | | |
| w/ R1 Distill | 90% | $0.7167_{(+3.30\%)}$ | $0.6900_{(-0.55\%)}$ | $0.6933_{(-0.07\%)}$ | $0.7100_{(+2.34\%)}$ | $0.6867_{(-1.02\%)}$ | $0.6993_{(+0.79\%)}$ |
| | 80% | $0.6867_{(-1.02\%)}$ | $0.6933_{(-0.07\%)}$ | $0.6933_{(-0.07\%)}$ | $0.7067_{(+1.86\%)}$ | $0.6767_{(-2.47\%)}$ | $0.6913_{(-0.36\%)}$ |
| | 70% | $0.6933_{(-0.07\%)}$ | $0.6633_{(-4.40\%)}$ | $0.7100_{(+2.34\%)}$ | $0.7100_{(+2.34\%)}$ | $0.7000_{(+0.89\%)}$ | $0.6953_{(+0.22\%)}$ |
| | 60% | $0.6833_{(-1.51\%)}$ | $0.6900_{(-0.55\%)}$ | $0.6900_{(-0.55\%)}$ | $0.7133_{(+2.81\%)}$ | $0.7067_{(+1.86\%)}$ | $0.6967_{(+0.42\%)}$ |
| | 50% | $0.6700_{(-3.43\%)}$ | $0.6967_{(+0.42\%)}$ | $0.7067_{(+1.86\%)}$ | $0.7000_{(+0.89\%)}$ | $0.6867_{(-1.02\%)}$ | $0.6920_{(-0.26\%)}$ |
| | 40% | $0.6767_{(-2.47\%)}$ | $0.6800_{(-1.99\%)}$ | $0.5967_{(-13.99\%)}$ | $0.6967_{(+0.42\%)}$ | $0.7133_{(+2.81\%)}$ | $0.6727_{(-3.04\%)}$ |
| | 30% | $0.6600_{(-4.87\%)}$ | $0.5900_{(-14.96\%)}$ | $0.5833_{(-15.93\%)}$ | $0.6700_{(-3.43\%)}$ | $0.7000_{(+0.89\%)}$ | $0.6407_{(-7.66\%)}$ |
| | 20% | $0.6200_{(-10.64\%)}$ | $0.4933_{(-28.90\%)}$ | $0.5633_{(-18.81\%)}$ | $0.6833_{(-1.51\%)}$ | $0.6533_{(-5.84\%)}$ | $0.6026_{(-13.14\%)}$ |
| | 10% | $0.5167_{(-25.53\%)}$ | $0.5567_{(-19.76\%)}$ | $0.5767_{(-16.88\%)}$ | $0.6267_{(-9.67\%)}$ | $0.6433_{(-7.28\%)}$ | $0.5840_{(-15.82\%)}$ |
| | Baseline | | | FullKV: 0.5122 | | | |
| w/o Instruct Tuning | 90% | $0.5061_{(-1.20\%)}$ | $0.5183_{(+1.20\%)}$ | $0.5122_{(+0.00\%)}$ | $0.5122_{(+0.00\%)}$ | $0.5122_{(+0.00\%)}$ | $0.5122_{(+0.00\%)}$ |
| | 80% | $0.5061_{(-1.20\%)}$ | $0.5183_{(+1.20\%)}$ | $0.5183_{(+1.20\%)}$ | $0.5305_{(+3.60\%)}$ | $0.5061_{(-1.20\%)}$ | $0.5159_{(+0.70\%)}$ |
| | 70% | $0.5000_{(-2.40\%)}$ | $0.5244_{(+2.40\%)}$ | $0.5122_{(+0.00\%)}$ | $0.5183_{(+1.20\%)}$ | $0.5122_{(+0.00\%)}$ | $0.5134_{(+0.20\%)}$ |
| | 60% | $0.5061_{(-1.20\%)}$ | $0.5366_{(+4.80\%)}$ | $0.5366_{(+4.80\%)}$ | $0.5305_{(+3.60\%)}$ | $0.5244_{(+2.40\%)}$ | $0.5268_{(+2.90\%)}$ |
| | 50% | $0.4939_{(-3.60\%)}$ | $0.5427_{(+6.00\%)}$ | $0.5061_{(-1.20\%)}$ | $0.4939_{(-3.60\%)}$ | $0.4878_{(-4.80\%)}$ | $0.5049_{(-1.40\%)}$ |
| | 40% | $0.4817_{(-6.00\%)}$ | $0.5427_{(+6.00\%)}$ | $0.5244_{(+2.40\%)}$ | $0.4939_{(-3.60\%)}$ | $0.5000_{(-2.40\%)}$ | $0.5085_{(-0.70\%)}$ |
| | 30% | $0.4817_{(-6.00\%)}$ | $0.5305_{(+3.60\%)}$ | $0.5000_{(-2.40\%)}$ | $0.4939_{(-3.60\%)}$ | $0.4817_{(-6.00\%)}$ | $0.4976_{(-2.90\%)}$ |
| | 20% | $0.4634_{(-9.50\%)}$ | $0.5061_{(-1.20\%)}$ | $0.4939_{(-3.60\%)}$ | $0.4695_{(-8.30\%)}$ | $0.4878_{(-4.80\%)}$ | $0.4841_{(-5.50\%)}$ |
| | 10% | $0.3659_{(-28.60\%)}$ | $0.4634_{(-9.50\%)}$ | $0.4268_{(-16.70\%)}$ | $0.4207_{(-17.90\%)}$ | $0.4451_{(-13.10\%)}$ | $0.4244_{(-17.20\%)}$ |

*Table 8.* KV Cache Compression Performance Comparison on *Arithmetic Reasoning* with Different Instruction TuningSettings

LLaMA-3.1-8B-Instruct, fix the compression methods (ChunkKV and H2O), and apply identical compression ratios (10% and 30%) to two task families: a retrieval-oriented long-context workload (LongBench) and a high-density reasoning workload (many-shot GSM8K, prompt ≈4K tokens). Table 11 shows that LongBench-style retrieval remains comparatively stable, while many-shot GSM8K degrades sharply at 10%—supporting our framing that retrieval-oriented benchmarks underestimate the impact of compression on reasoning.

### D.3. Ablation: Prefill-only vs. ShotKV

To assess the contribution of the decoding-phase compression, we ablate it by retaining only prefill compression and removing dynamic decoding-side compression. The ablation is run on two long-context settings.

**LG-GSM8K (LLaMA-3.1-8B-Instruct).** As summarized in Table 12, the prefill-only variant substantially underperforms full ShotKV across all compression ratios, confirming the importance of the prefill–decoding separation.

**Many-shot GSM8K (LLaMA-3.1-8B-Instruct).** The same pattern holds on many-shot arithmetic reasoning (Table 13): removing the decoding-side compression component reduces accuracy by ∼2–3 absolute points across ratios, with the gap widest under aggressive (10%) compression.

### D.4. Shot ordering and exemplar robustness

To probe whether ShotKV's attention-based scoring is robust to superficial perturbations of the prompt, we run two controlled variations on many-shot GSM8K with LLaMA-3.1-8B-Instruct: *(i) Re-ordered*, where we shuffle the order of the few-shot exemplars while keeping the set fixed; and *(ii) Diff-Example*, where we replace the exemplar set with a different sample of GSM8K problems of the same size. Table 14 shows that ShotKV is broadly stable to shot reordering and reasonably stable

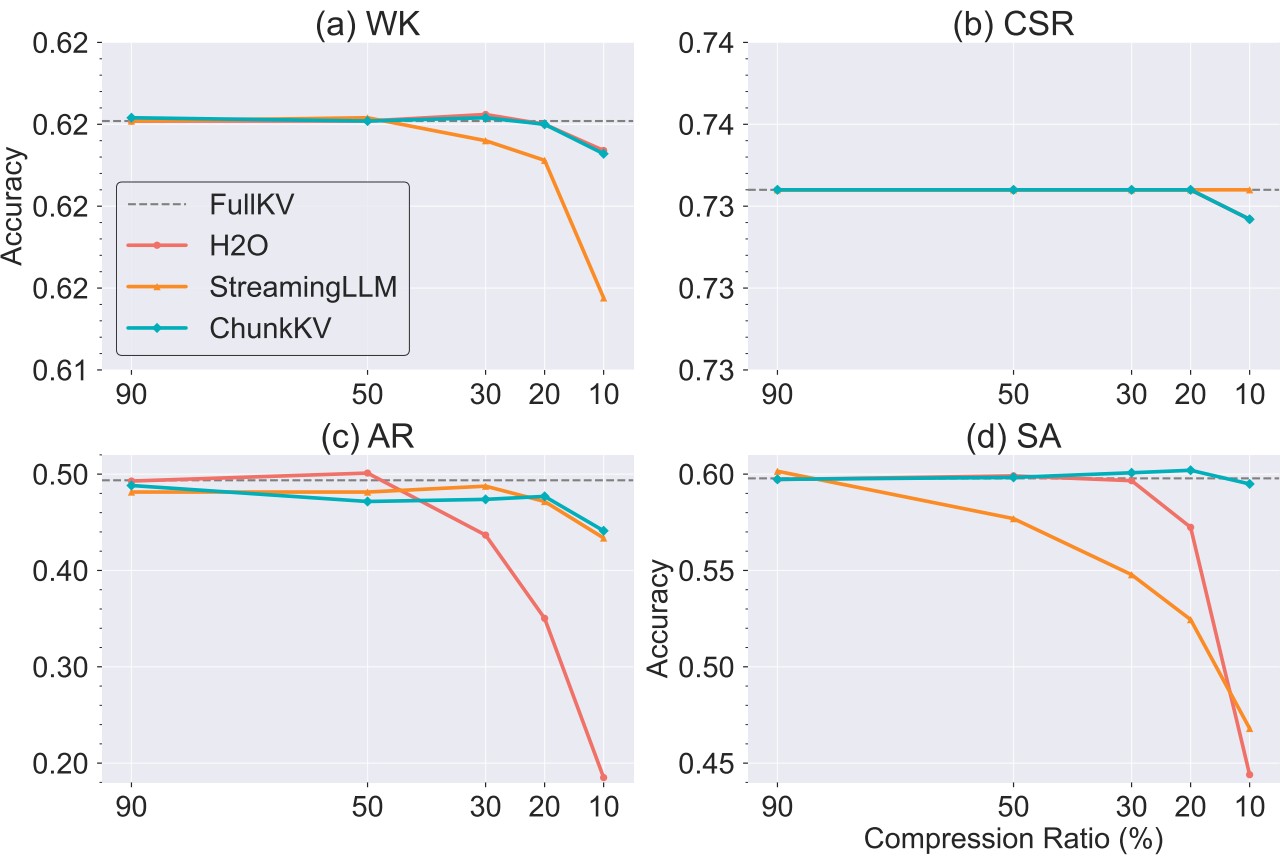

*Figure 9.* Performance Comparison of KV Cache Compression Methods Across Tasks with Mistral-7B-Instruct.

across exemplar choices: variations are within ~1.0 absolute point across all compression ratios. We note that this analysis demonstrates robustness to ordering and exemplar replacement; a fully principled validation of the score under arbitrary semantic corruption remains future work.

### D.5. Extended efficiency comparison

Table 5 in the main paper compares end-to-end efficiency of ShotKV, FullKV, and two representative compression baselines at the 4K–4K setting. For completeness, Table 15 reports the same configuration on the A40 server with batch size 1 across all four methods. ShotKV achieves efficiency comparable to ChunkKV and SnapKV; the design objective is not a radically different speed profile, but rather to preserve reasoning-relevant semantic structure under similar efficiency budgets.

### D.6. More experiments on other models

To further validate the generality of our findings, we evaluate the impact of KV cache compression on additional model families beyond LLaMA-3.1.

**Mistral-7B-Instruct.**   As shown in Figure 9, we observe that various KV cache compression methods lead to significant performance degradation across multiple fundamental tasks, especially under aggressive compression ratios. This result demonstrates that the reduction in foundation abilities due to KV cache compression is not limited to a specific model family, but is a general phenomenon affecting different LLM architectures.

**Qwen3-4B-Base.**   We evaluate Qwen3-4B-Base on KVFundaBench to test cross-family generality. Table 16 reports performance on Arithmetic Reasoning (AR) and World Knowledge (WK) at 10% and 30% compression with two representative methods. The same pattern observed on the LLaMA and Mistral families holds on Qwen3: aggressive token-level

---

**Algorithm 1** ShotKV: Shot-aware KV Cache Compression

---

**Require:** Prompt with $n$ shots $\{s_1, ..., s_n\}$, prefill ratio $r_p$, decoding ratio $r_d$, number of layers $L$
**Ensure:** Compressed KV cache $KV_{\text{total}}$

1: Initialize $KV_{\text{total}} \leftarrow \emptyset$
2: **for** each layer $l \in \{1, ..., L\}$ **do**
3:     // Phase 1: Prefill Compression (Layer-wise)
4:     **for** each shot $s_i$ in $\{s_1, ..., s_n\}$ **do**
5:         Compute $\text{Score}^l_{\text{prefill}}(s_i) = \frac{1}{k_i} \sum_{t \in s_i} \sum_{h=1}^{H} \alpha^l_{t,h}$
6:     **end for**
7:     Sort shots by $\text{Score}^l_{\text{prefill}}(s_i)$ in descending order
8:     $S_{\text{preserved},l} \leftarrow$ Select shots until $\sum_{s_i} k_i \leq r_p \times |KV_{\text{prefill}}|$
9:     $KV^C_{\text{prefill},l} \leftarrow \text{Compress}(\{s_i | s_i \in S_{\text{preserved},l}\})$
10:     // Phase 2: Decoding Compression (Layer-wise, dynamic)
11:     **for** each decoding step **do**
12:         **for** each token $t$ in $KV_{\text{decoding}}$ **do**
13:             Compute $\text{Score}^l_{\text{decoding}}(t) = \sum_{h=1}^{H} \alpha^l_{t,h}$
14:         **end for**
15:         $k \leftarrow r_d \times |KV_{\text{decoding}}|$
16:         $KV^C_{\text{decoding},l} \leftarrow \text{TopK}(KV_{\text{decoding}}, \text{Score}^l_{\text{decoding}}, k)$
17:     **end for**
18:     $KV_{\text{total},l} \leftarrow KV^C_{\text{prefill},l} \cup KV^C_{\text{decoding},l}$
19:     $KV_{\text{total}} \leftarrow KV_{\text{total}} \cup \{KV_{\text{total},l}\}$
20: **end for**
21: **return** $KV_{\text{total}}$

---

compression (H2O at 10%) causes a substantial collapse on AR ($87.79 \rightarrow 49.20$), while WK remains comparatively robust; chunk-level retention (ChunkKV) significantly mitigates the AR degradation ($87.79 \rightarrow 64.41$ at 10%). Together with the LLaMA-3.1 / Mistral / DeepSeek-R1 results elsewhere in the paper, this provides cross-family evidence for the High-Density-Reasoning vulnerability identified by KVFundaBench. The full ShotKV pipeline on Qwen3 follows the same trend; we report the complete sweep below as it became available.

## E. ShotKV Pseudocode and Algorithmic Details

This section provides the detailed algorithmic description of ShotKV.

### E.1. Pseudocode

The detailed algorithm of ShotKV is presented in Algorithm 1. Our method consists of two main phases: prefill compression and decoding compression. During the prefill phase, we compute an importance score for each shot by averaging the attention weights across all tokens, heads, and layers within that shot. This score $\text{Score}_{\text{prefill}}(s_i)$ is normalized by the shot length $k_i$ to avoid bias towards longer shots. Shots are then sorted by their scores and preserved until reaching the specified prefill ratio $r_p$.

In the decoding phase, compression is performed dynamically at each step. For each token in the decoding KV cache, we calculate its importance score $\text{Score}_{\text{decoding}}(t)$ by summing attention weights across all heads and layers. The top-k tokens are retained based on the decoding ratio $r_d$. Finally, the compressed KV cache is formed by combining both the preserved prefill and decoding caches.

This two-phase approach allows for different compression strategies during prefill and decoding stages, recognizing their distinct roles in the inference process. The shot-aware design during prefill ensures that the most informative examples are preserved, while the token-level compression during decoding maintains essential recent context.

# F. Evaluation Benchmark

## F.1. Dataset Details

Detailed statistics for each benchmark dataset are provided in Table 17. For HotpotQA, we only report results under the 10% compression ratio using the LLaMA-3-8B-Instruct model.

The hyper-parameters for different observations are shown in Table 18. The temperature for the experiments are set to 0 for ensuring the deterministic results.

*Table 9.* Performance Comparison of Different Shot Numbers on GSM8K

| Shot | Ratio | StreamingLLM | H2O | SnapKV | PyramidKV | ChunkKV | Average ↑ |
|---|---|---|---|---|---|---|---|
| | Baseline | | | FullKV: 0.7149 | | | |
| 1-shot | 90% | $0.7013_{(-1.90\%)}$ | $0.7172_{(+0.30\%)}$ | $0.7142_{(-0.10\%)}$ | $0.7020_{(-1.80\%)}$ | $0.7172_{(+0.30\%)}$ | $0.7104_{(-0.60\%)}$ |
| | 80% | $0.6892_{(-3.60\%)}$ | $0.7089_{(-0.80\%)}$ | $0.7066_{(-1.20\%)}$ | $0.6952_{(-2.80\%)}$ | $0.7081_{(-1.00\%)}$ | $0.7016_{(-1.90\%)}$ |
| | 70% | $0.6816_{(-4.70\%)}$ | $0.6914_{(-3.30\%)}$ | $0.6945_{(-2.90\%)}$ | $0.6884_{(-3.70\%)}$ | $0.7127_{(-0.30\%)}$ | $0.6937_{(-3.00\%)}$ |
| | 60% | $0.6884_{(-3.70\%)}$ | $0.6831_{(-4.40\%)}$ | $0.6914_{(-3.30\%)}$ | $0.6816_{(-4.70\%)}$ | $0.6990_{(-2.20\%)}$ | $0.6887_{(-3.70\%)}$ |
| | 50% | $0.6952_{(-2.80\%)}$ | $0.6596_{(-7.70\%)}$ | $0.6611_{(-7.50\%)}$ | $0.6717_{(-6.00\%)}$ | $0.6732_{(-5.80\%)}$ | $0.6722_{(-6.00\%)}$ |
| | 40% | $0.6657_{(-6.90\%)}$ | $0.6202_{(-13.20\%)}$ | $0.6065_{(-15.20\%)}$ | $0.6475_{(-9.40\%)}$ | $0.6050_{(-15.40\%)}$ | $0.6290_{(-12.00\%)}$ |
| | 30% | $0.5118_{(-28.40\%)}$ | $0.5004_{(-30.00\%)}$ | $0.5042_{(-29.50\%)}$ | $0.5898_{(-17.50\%)}$ | $0.4011_{(-43.90\%)}$ | $0.5015_{(-29.90\%)}$ |
| | 20% | $0.2320_{(-67.50\%)}$ | $0.2714_{(-62.00\%)}$ | $0.2654_{(-62.90\%)}$ | $0.3973_{(-44.40\%)}$ | $0.1319_{(-81.60\%)}$ | $0.2596_{(-63.70\%)}$ |
| | 10% | $0.0296_{(-95.90\%)}$ | $0.0243_{(-96.60\%)}$ | $0.0296_{(-95.90\%)}$ | $0.1236_{(-82.70\%)}$ | $0.0190_{(-97.30\%)}$ | $0.0452_{(-93.70\%)}$ |
| | Baseline | | | FullKV: 0.7574 | | | |
| 2-shot | 90% | $0.7544_{(-0.40\%)}$ | $0.7604_{(+0.40\%)}$ | $0.7574_{(+0.00\%)}$ | $0.7612_{(+0.50\%)}$ | $0.7627_{(+0.70\%)}$ | $0.7592_{(+0.20\%)}$ |
| | 80% | $0.7551_{(-0.30\%)}$ | $0.7521_{(-0.70\%)}$ | $0.7559_{(-0.20\%)}$ | $0.7559_{(-0.20\%)}$ | $0.7589_{(+0.20\%)}$ | $0.7556_{(-0.20\%)}$ |
| | 70% | $0.7521_{(-0.70\%)}$ | $0.7453_{(-1.60\%)}$ | $0.7566_{(-0.10\%)}$ | $0.7574_{(+0.00\%)}$ | $0.7642_{(+0.90\%)}$ | $0.7551_{(-0.30\%)}$ |
| | 60% | $0.7475_{(-1.30\%)}$ | $0.7506_{(-0.90\%)}$ | $0.7521_{(-0.70\%)}$ | $0.7589_{(+0.20\%)}$ | $0.7695_{(+1.60\%)}$ | $0.7557_{(-0.20\%)}$ |
| | 50% | $0.7460_{(-1.50\%)}$ | $0.7437_{(-1.80\%)}$ | $0.7437_{(-1.80\%)}$ | $0.7604_{(+0.40\%)}$ | $0.7619_{(+0.60\%)}$ | $0.7511_{(-0.80\%)}$ |
| | 40% | $0.7445_{(-1.70\%)}$ | $0.7081_{(-6.50\%)}$ | $0.7202_{(-4.90\%)}$ | $0.7309_{(-3.50\%)}$ | $0.7650_{(+1.00\%)}$ | $0.7337_{(-3.10\%)}$ |
| | 30% | $0.7506_{(-0.90\%)}$ | $0.6133_{(-19.00\%)}$ | $0.6657_{(-12.10\%)}$ | $0.7036_{(-7.10\%)}$ | $0.7445_{(-1.70\%)}$ | $0.6955_{(-8.20\%)}$ |
| | 20% | $0.6217_{(-17.90\%)}$ | $0.4412_{(-41.70\%)}$ | $0.4936_{(-34.80\%)}$ | $0.5534_{(-26.90\%)}$ | $0.5368_{(-29.10\%)}$ | $0.5293_{(-30.10\%)}$ |
| | 10% | $0.1516_{(-80.00\%)}$ | $0.1759_{(-76.80\%)}$ | $0.1622_{(-78.60\%)}$ | $0.2244_{(-70.40\%)}$ | $0.0735_{(-90.30\%)}$ | $0.1575_{(-79.20\%)}$ |
| | Baseline | | | FullKV: 0.7597 | | | |
| 4-shot | 90% | $0.7597_{(+0.00\%)}$ | $0.7604_{(+0.10\%)}$ | $0.7650_{(+0.70\%)}$ | $0.7642_{(+0.60\%)}$ | $0.7657_{(+0.80\%)}$ | $0.7630_{(+0.40\%)}$ |
| | 80% | $0.7559_{(-0.50\%)}$ | $0.7688_{(+1.20\%)}$ | $0.7695_{(+1.30\%)}$ | $0.7680_{(+1.10\%)}$ | $0.7642_{(+0.60\%)}$ | $0.7653_{(+0.70\%)}$ |
| | 70% | $0.7597_{(+0.00\%)}$ | $0.7695_{(+1.30\%)}$ | $0.7680_{(+1.10\%)}$ | $0.7710_{(+1.50\%)}$ | $0.7726_{(+1.70\%)}$ | $0.7682_{(+1.10\%)}$ |
| | 60% | $0.7369_{(-3.00\%)}$ | $0.7726_{(+1.70\%)}$ | $0.7688_{(+1.20\%)}$ | $0.7635_{(+0.50\%)}$ | $0.7718_{(+1.60\%)}$ | $0.7627_{(+0.40\%)}$ |
| | 50% | $0.7475_{(-1.60\%)}$ | $0.7612_{(+0.20\%)}$ | $0.7619_{(+0.30\%)}$ | $0.7665_{(+0.90\%)}$ | $0.7635_{(+0.50\%)}$ | $0.7601_{(+0.10\%)}$ |
| | 40% | $0.7165_{(-5.70\%)}$ | $0.7339_{(-3.40\%)}$ | $0.7377_{(-2.90\%)}$ | $0.7483_{(-1.50\%)}$ | $0.7612_{(+0.20\%)}$ | $0.7395_{(-2.70\%)}$ |
| | 30% | $0.6558_{(-13.70\%)}$ | $0.6603_{(-13.10\%)}$ | $0.7111_{(-6.40\%)}$ | $0.7263_{(-4.40\%)}$ | $0.7597_{(+0.00\%)}$ | $0.7026_{(-7.50\%)}$ |
| | 20% | $0.6224_{(-18.10\%)}$ | $0.5625_{(-26.00\%)}$ | $0.6065_{(-20.20\%)}$ | $0.6543_{(-13.90\%)}$ | $0.7468_{(-1.70\%)}$ | $0.6385_{(-16.00\%)}$ |
| | 10% | $0.4708_{(-38.00\%)}$ | $0.3980_{(-47.60\%)}$ | $0.3995_{(-47.40\%)}$ | $0.4321_{(-43.10\%)}$ | $0.3434_{(-54.80\%)}$ | $0.4088_{(-46.20\%)}$ |
| | Baseline | | | FullKV: 0.7680 | | | |
| 6-shot | 90% | $0.7551_{(-1.70\%)}$ | $0.7748_{(+0.90\%)}$ | $0.7839_{(+2.10\%)}$ | $0.7794_{(+1.50\%)}$ | $0.7794_{(+1.50\%)}$ | $0.7745_{(+0.90\%)}$ |
| | 80% | $0.7642_{(-0.50\%)}$ | $0.7756_{(+1.00\%)}$ | $0.7809_{(+1.70\%)}$ | $0.7741_{(+0.80\%)}$ | $0.7786_{(+1.40\%)}$ | $0.7747_{(+0.90\%)}$ |
| | 70% | $0.7513_{(-2.20\%)}$ | $0.7771_{(+1.20\%)}$ | $0.7809_{(+1.70\%)}$ | $0.7771_{(+1.20\%)}$ | $0.7786_{(+1.40\%)}$ | $0.7730_{(+0.70\%)}$ |
| | 60% | $0.7468_{(-2.80\%)}$ | $0.7748_{(+0.90\%)}$ | $0.7733_{(+0.70\%)}$ | $0.7771_{(+1.20\%)}$ | $0.7809_{(+1.70\%)}$ | $0.7706_{(+0.30\%)}$ |
| | 50% | $0.7407_{(-3.60\%)}$ | $0.7718_{(+0.50\%)}$ | $0.7718_{(+0.50\%)}$ | $0.7771_{(+1.20\%)}$ | $0.7718_{(+0.50\%)}$ | $0.7666_{(-0.20\%)}$ |
| | 40% | $0.7377_{(-3.90\%)}$ | $0.7506_{(-2.30\%)}$ | $0.7771_{(+1.20\%)}$ | $0.7688_{(+0.10\%)}$ | $0.7854_{(+2.30\%)}$ | $0.7639_{(-0.50\%)}$ |
| | 30% | $0.7058_{(-8.10\%)}$ | $0.7255_{(-5.50\%)}$ | $0.7392_{(-3.70\%)}$ | $0.7491_{(-2.50\%)}$ | $0.7763_{(+1.10\%)}$ | $0.7392_{(-3.70\%)}$ |
| | 20% | $0.5921_{(-22.90\%)}$ | $0.6232_{(-18.80\%)}$ | $0.6732_{(-12.30\%)}$ | $0.6960_{(-9.40\%)}$ | $0.7665_{(-0.20\%)}$ | $0.6702_{(-12.70\%)}$ |
| | 10% | $0.4572_{(-40.50\%)}$ | $0.4481_{(-41.60\%)}$ | $0.4958_{(-35.40\%)}$ | $0.4458_{(-41.90\%)}$ | $0.5565_{(-27.50\%)}$ | $0.4807_{(-37.40\%)}$ |
| | Baseline | | | FullKV: 0.7945 | | | |
| 8-shot | 90% | $0.7695_{(-3.10\%)}$ | $0.7923_{(-0.30\%)}$ | $0.7839_{(-1.30\%)}$ | $0.7854_{(-1.10\%)}$ | $0.7824_{(-1.50\%)}$ | $0.7827_{(-1.50\%)}$ |
| | 80% | $0.7642_{(-3.80\%)}$ | $0.7938_{(-0.10\%)}$ | $0.7824_{(-1.50\%)}$ | $0.7900_{(-0.60\%)}$ | $0.7824_{(-1.50\%)}$ | $0.7826_{(-1.50\%)}$ |
| | 70% | $0.7642_{(-3.80\%)}$ | $0.7900_{(-0.60\%)}$ | $0.7923_{(-0.30\%)}$ | $0.7983_{(+0.50\%)}$ | $0.7809_{(-1.70\%)}$ | $0.7851_{(-1.20\%)}$ |
| | 60% | $0.7650_{(-3.70\%)}$ | $0.7809_{(-1.70\%)}$ | $0.7885_{(-0.80\%)}$ | $0.7923_{(-0.30\%)}$ | $0.7885_{(-0.80\%)}$ | $0.7830_{(-1.50\%)}$ |
| | 50% | $0.7657_{(-3.60\%)}$ | $0.7854_{(-1.10\%)}$ | $0.7847_{(-1.20\%)}$ | $0.7854_{(-1.10\%)}$ | $0.7824_{(-1.50\%)}$ | $0.7807_{(-1.70\%)}$ |
| | 40% | $0.7491_{(-5.70\%)}$ | $0.7688_{(-3.20\%)}$ | $0.7756_{(-2.40\%)}$ | $0.7839_{(-1.30\%)}$ | $0.7763_{(-2.30\%)}$ | $0.7707_{(-3.00\%)}$ |
| | 30% | $0.7051_{(-11.20\%)}$ | $0.7225_{(-9.10\%)}$ | $0.7619_{(-4.10\%)}$ | $0.7718_{(-2.90\%)}$ | $0.7733_{(-2.70\%)}$ | $0.7469_{(-6.00\%)}$ |
| | 20% | $0.6384_{(-19.70\%)}$ | $0.6406_{(-19.40\%)}$ | $0.6884_{(-13.40\%)}$ | $0.7142_{(-10.10\%)}$ | $0.7763_{(-2.30\%)}$ | $0.6916_{(-13.00\%)}$ |
| | 10% | $0.4784_{(-39.80\%)}$ | $0.4503_{(-43.30\%)}$ | $0.5034_{(-36.60\%)}$ | $0.4829_{(-39.20\%)}$ | $0.6566_{(-17.40\%)}$ | $0.5143_{(-35.30\%)}$ |

*Table 10.* Performance Comparison of Different KV Cache Compression Methods on Many-shot GSM8K

| Benchmark | Ratio | StreamingLLM | H2O | SnapKV | PyramidKV | ChunkKV | Average ↑ |
|---|---|---|---|---|---|---|---|
| | Baseline | LLaMA-3.1-8B-Instruct FullKV: 0.8235 | | | | | |
| | 90% | $0.7728_{(-6.16\%)}$ | $0.8142_{(-1.13\%)}$ | $0.8137_{(-1.19\%)}$ | $0.7932_{(-3.68\%)}$ | $0.8233_{(-0.02\%)}$ | $0.8034_{(-2.44\%)}$ |
| | 80% | $0.7935_{(-3.64\%)}$ | $0.8334_{(+1.20\%)}$ | $0.8138_{(-1.18\%)}$ | $0.8037_{(-2.40\%)}$ | $0.7932_{(-3.68\%)}$ | $0.8075_{(-1.94\%)}$ |
| | 70% | $0.8038_{(-2.39\%)}$ | $0.8136_{(-1.20\%)}$ | $0.7832_{(-4.89\%)}$ | $0.7932_{(-3.68\%)}$ | $0.8037_{(-2.40\%)}$ | $0.7995_{(-2.91\%)}$ |
| | 60% | $0.7932_{(-3.68\%)}$ | $0.8142_{(-1.13\%)}$ | $0.8037_{(-2.40\%)}$ | $0.7935_{(-3.64\%)}$ | $0.8038_{(-2.39\%)}$ | $0.8017_{(-2.65\%)}$ |
| | 50% | $0.7934_{(-3.65\%)}$ | $0.8137_{(-1.19\%)}$ | $0.7932_{(-3.68\%)}$ | $0.7932_{(-3.68\%)}$ | $0.7835_{(-4.86\%)}$ | $0.7954_{(-3.41\%)}$ |
| | 40% | $0.8037_{(-2.40\%)}$ | $0.7832_{(-4.89\%)}$ | $0.7935_{(-3.64\%)}$ | $0.7834_{(-4.87\%)}$ | $0.7832_{(-4.89\%)}$ | $0.7894_{(-4.14\%)}$ |
| | 30% | $0.7835_{(-4.86\%)}$ | $0.7932_{(-3.68\%)}$ | $0.8038_{(-2.39\%)}$ | $0.7934_{(-3.65\%)}$ | $0.7932_{(-3.68\%)}$ | $0.7934_{(-3.65\%)}$ |
| | 20% | $0.7537_{(-8.47\%)}$ | $0.7428_{(-9.80\%)}$ | $0.7934_{(-3.65\%)}$ | $0.7832_{(-4.89\%)}$ | $0.7835_{(-4.86\%)}$ | $0.7713_{(-6.34\%)}$ |
| | 10% | $0.7432_{(-9.75\%)}$ | $0.5127_{(-37.74\%)}$ | $0.6827_{(-17.10\%)}$ | $0.7037_{(-14.55\%)}$ | $0.7932_{(-3.68\%)}$ | $0.6871_{(-16.56\%)}$ |
| Many-shot | Baseline | R1-Distill-Llama-8B FullKV: 0.7123 | | | | | |
| GSM8K | 90% | $0.7123_{(+1.42\%)}$ | $0.6612_{(-5.85\%)}$ | $0.6534_{(-6.96\%)}$ | $0.6912_{(-1.58\%)}$ | $0.6923_{(-1.42\%)}$ | $0.6821_{(-2.88\%)}$ |
| | 80% | $0.7234_{(+3.00\%)}$ | $0.6534_{(-6.96\%)}$ | $0.7123_{(+1.42\%)}$ | $0.6423_{(-8.54\%)}$ | $0.7123_{(+1.42\%)}$ | $0.6887_{(-1.94\%)}$ |
| | 70% | $0.7412_{(+5.54\%)}$ | $0.6523_{(-7.12\%)}$ | $0.7234_{(+3.00\%)}$ | $0.6923_{(-1.42\%)}$ | $0.7234_{(+3.00\%)}$ | $0.7065_{(+0.60\%)}$ |
| | 60% | $0.7423_{(+5.69\%)}$ | $0.6912_{(-1.58\%)}$ | $0.6912_{(-1.58\%)}$ | $0.6823_{(-2.85\%)}$ | $0.6634_{(-5.54\%)}$ | $0.6941_{(-1.17\%)}$ |
| | 50% | $0.7234_{(+3.00\%)}$ | $0.7134_{(+1.58\%)}$ | $0.7312_{(+4.12\%)}$ | $0.7123_{(+1.42\%)}$ | $0.7123_{(+1.42\%)}$ | $0.7185_{(+2.31\%)}$ |
| | 40% | $0.7123_{(+1.42\%)}$ | $0.6923_{(-1.42\%)}$ | $0.6923_{(-1.42\%)}$ | $0.7023_{(+0.00\%)}$ | $0.7234_{(+3.00\%)}$ | $0.7045_{(+0.31\%)}$ |
| | 30% | $0.6523_{(-7.12\%)}$ | $0.7312_{(+4.12\%)}$ | $0.6634_{(-5.54\%)}$ | $0.7423_{(+5.69\%)}$ | $0.6912_{(-1.58\%)}$ | $0.6961_{(-0.88\%)}$ |
| | 20% | $0.6912_{(-1.58\%)}$ | $0.5834_{(-16.93\%)}$ | $0.5123_{(-27.05\%)}$ | $0.6823_{(-2.85\%)}$ | $0.6634_{(-5.54\%)}$ | $0.6265_{(-10.79\%)}$ |
| | 10% | $0.6323_{(-9.97\%)}$ | $0.5423_{(-22.78\%)}$ | $0.5412_{(-22.94\%)}$ | $0.5923_{(-15.66\%)}$ | $0.6823_{(-2.85\%)}$ | $0.5981_{(-14.84\%)}$ |

*Table 11.* Matched-setting comparison on LLaMA-3.1-8B-Instruct. Same model, methods, and compression ratios applied to retrieval-oriented (LongBench) vs. high-density reasoning (many-shot GSM8K). $\Delta$ is absolute drop from FullKV.

| Method | Task family | FullKV | 10% | | 30% | |
|---|---|---|---|---|---|---|
| | | | Score | $\Delta$ | Score | $\Delta$ |
| ChunkKV | Retrieval-oriented (LongBench) | 41.46 | 40.51 | $-0.95$ | 41.59 | $+0.13$ |
| ChunkKV | High-density reasoning (GSM8K) | 79.45 | 65.66 | $-13.79$ | 77.63 | $-1.82$ |
| H2O | Retrieval-oriented (LongBench) | 41.46 | 37.06 | $-4.40$ | 39.23 | $-2.23$ |
| H2O | High-density reasoning (GSM8K) | 79.45 | 45.03 | $-34.42$ | 64.06 | $-15.39$ |

*Table 12.* LLaMA-3.1-8B-Instruct on LG-GSM8K: ShotKV vs. Prefill-only.

| Method | 40% | 35% | 30% | 25% |
|---|---|---|---|---|
| ShotKV | 47.33 | 41.33 | 38.33 | 26.83 |
| Prefill-only | 42.15 | 30.14 | 27.63 | 18.59 |

*Table 13.* LLaMA-3.1-8B-Instruct on many-shot GSM8K: ShotKV vs. Prefill-only.

| Method | 40% | 30% | 20% | 10% |
|---|---|---|---|---|
| ShotKV | 81.07 | 80.82 | 80.57 | 80.37 |
| Prefill-only | 79.07 | 78.82 | 78.57 | 77.26 |

*Table 14.* Shot-ordering and exemplar-choice robustness of ShotKV on many-shot GSM8K (LLaMA-3.1-8B-Instruct).

| Method | 40% | 30% | 20% | 10% |
|---|---|---|---|---|
| ShotKV | 81.07 | 80.82 | 80.57 | 80.37 |
| ShotKV (Re-ordered 1) | 80.82 | 80.57 | 80.34 | 80.15 |
| ShotKV (Re-ordered 2) | 81.12 | 80.87 | 80.62 | 80.22 |
| ShotKV (Diff-Example 1) | 80.87 | 80.14 | 79.96 | 79.72 |
| ShotKV (Diff-Example 2) | 80.97 | 80.94 | 80.91 | 80.88 |

*Table 15.* Extended efficiency comparison on A40, batch size 1, input/output 4096/4096. Percentages denote relative gain over FullKV.

| Method | Latency (s) ↓ | Throughput (T/S) ↑ |
|---|---|---|
| FullKV | 175.50 | 37.73 |
| ChunkKV | 160.32 (8.6%) | 41.30 (9.5%) |
| SnapKV | 163.45 (6.9%) | 40.51 (7.4%) |
| ShotKV | 162.85 (7.2%) | 41.12 (9.0%) |

*Table 16.* Qwen3-4B-Base on KVFundaBench: AR (Arithmetic Reasoning) and WK (World Knowledge) at 10% and 30% compression.

| Task | FullKV | H2O (token-level) | | ChunkKV (chunk-level) | |
|---|---|---|---|---|---|
| | | 10% | 30% | 10% | 30% |
| AR | 87.79 | 49.20 | 73.78 | 64.41 | 80.41 |
| WK | 72.99 | 72.03 | 71.23 | 72.99 | 72.31 |

*Table 17.* The statistics of the datasets used in this paper. # TEST denote the number of training data and test data, respectively.

| DATASET | TASK TYPE | # TEST | METRIC | EVALUATION METHOD |
|---|---|---|---|---|
| MMLU (Hendrycks et al., 2021a) | World Knowledge | 14,079 | Accuracy | Generation-Based |
| GSM8K (Cobbe et al., 2021) | Arithmetic | 1,319 | Exact match | Generation-Based |
| CSQA (Talmor et al., 2019) | Commonsense | 1,221 | Accuracy | Generation-Based |
| HumanEval (Chen et al., 2021) | Code Generation | 164 | Pass@1 rate | Generation-Based |
| JailBreakV (Luo et al., 2024) | Safety | 28,000 | Attack success rate | Generation-Based |
| HotpotQA (Yang et al., 2018) | Document QA (Multi-hop) | 7,405 | Accuracy | Generation-Based |
| LongGenBench (Liu et al., 2024d) | Long-Context Generation | 23,000 | Accuracy | Generation-Based |

*Table 18.* Hyperparameters for Different Observations

| Benchmarks | Obs 1 | Obs 2 | Obs 3 | Obs 4 | Obs 5 | Obs 6 | |
|---|---|---|---|---|---|---|---|
| | Number of Shots | | | | | $K$ | $T$ |
| MMLU (Hendrycks et al., 2021a) | 5 | 5 | - | - | 0,5 | - | - |
| CommonsenseQA (Talmor et al., 2019) | 4 | 4 | - | - | - | - | - |
| GSM8K (Cobbe et al., 2021) | 8 | 8 | 1-8 | 50 | 0,8 | - | - |
| HumanEval (Chen et al., 2021) | 8 | 8 | - | - | - | - | - |
| JailBreakV (Luo et al., 2024) | 8 | 8 | - | - | - | - | - |
| LongGenBench-GSM8K (Liu et al., 2024d) | - | - | - | - | - | 35 | 20 |

