# OpenReview forum: "Semantic Integrity Matters: Benchmarking and Preserving High-Density Reasoning in KV Cache Compression"
_ICML.cc/2026/Conference — ICML 2026 regular_

### Official Review · Reviewer_h9vf · 2026-03-05

**Soundness:** 3
**Presentation:** 3
**Significance:** 2
**Originality:** 2
**Overall Recommendation:** 4
**Confidence:** 2

**Summary:**

This paper introduces the KVFundaBench and systematically reveals a key differentiated phenomenon in KV cache compression: while traditional retrieval tasks remain relatively robust under compression, “high-density reasoning” tasks involving chain-of-thought (CoT) coherence suffer from severe task-dependent performance degradation under aggressive compression due to the disruption of semantic reasoning chains. To mitigate this issue, the authors propose ShotKV, whose core idea is to treat the few-shot examples in prompts as indivisible semantic units and preserve them intact. Furthermore, the method adopts explicitly separated compression strategies for the prefill and decoding stages.

**Compliance With Llm Reviewing Policy:**

Affirmed.

**Final Justification:**

The authors have addressed my main concerns. However, I still hope that, as stated, they will reorganize the introduction and method sections in the final main paper to more clearly highlight the novelty of the approach, and also add a related work section.

**Key Questions For Authors:**

(1) In the Prefill Compression stage, ChunkKV performs sorting within the window, whereas the proposed method only performs sorting within the n-shot examples. Moreover, the method relies on Observation 4, which has already been demonstrated in ChunkKV. During the decoding phase, the proposed approach relies on attention-score-based ranking, which resembles an incremental extension of H2O. The authors should therefore more clearly articulate and emphasize the key innovations of their approach.

(2) The paper should include comparative experiments with KV quantization methods. In addition, ablation studies on prefill compression ratios and decoding compression ratios are currently missing.

(3) The proposed method appears to rely heavily on the presence of a clear few-shot structure in the prompt. For zero-shot settings or unstructured long-text scenarios (e.g., long-document summarization), the paper only provides preliminary experimental results. This raises concerns regarding the method’s applicability in practical deployments. Furthermore, while Table 4 presents results under different compression ratios, Table 3 only reports results at a 10% compression ratio, which limits the comprehensiveness of the evaluation.

(4) While I understand the page limit constraints, the main paper should still include a Related Work section that briefly discusses the most closely related studies, with additional details provided in the appendix if necessary. Moving the entire Related Work section to the appendix is not an appropriate practice.

I will raise my score if the author addresses my issue.

**Limitations:**

Yes

**Strengths And Weaknesses:**

Strength:

(1) The paper presents a clear motivation, and the experimental setup is well-designed.
(2) The analysis based on KVFundaBench is comprehensive.
(3) The experimental results demonstrate that the proposed method achieves significant improvements.

Weakness:

See the Key Questions for Authors.

---

> ### Author Rebuttal · Authors · 2026-03-28
>
> We sincerely thank the reviewer for recognizing the paper's clear motivation, comprehensive benchmark analysis, and strong empirical improvements, and for indicating that the score may improve if the concerns are addressed. We address your concerns below.
>
>
>
> ---
>
> **Weakness 1 – Novelty Relative to ChunkKV and H2O**
>
> ***Answer to Weakness 1:***
> We agree that the current draft should articulate the distinction more clearly. Our contribution is **not** that each component is individually unprecedented; rather, the novelty lies in the **task-motivated combination** of two design choices derived from the benchmark findings:
>
> **1. Shot-aware semantic unit preservation during prefill.** **ChunkKV** supports the observation that chunk-level retention is stronger than token-level retention on many-shot reasoning. **ShotKV** sharpens this idea by preserving **few-shot examples as explicit semantic units**, rather than applying generic fixed-window chunk ranking.
>
> **2. Explicit prefill/decoding separation.** Unlike static prefill-oriented methods, **ShotKV** combines shot-level prefill preservation with **dynamic decoding compression**, which is critical for long-context generation. This is empirically validated in **Table 11**, where the prefill-only ablation is substantially weaker than full **ShotKV**.
>
> **3. Benchmark-driven method design.** The paper's contribution is also the connection from **KVFundaBench observations** to a concrete design principle. The method is intentionally simple because it is an insight-driven response to the failure mode exposed by the benchmark.
>
> We will revise the method section to make this positioning explicit and avoid giving the impression that we claim novelty for generic attention-based ranking alone.
>
> ---
>
> **Weakness 2 – Missing Quantization / Ratio Ablation**
>
> We agree these are useful extensions. Our current study focuses on **KV compression by token retention / eviction**, so the baselines are drawn from that line of work; we will clarify this scope more explicitly. For ablation, we have added a direct experiment on **LLaMA-3.1-8B-Instruct** that isolates the contribution of decoding-phase compression. On **many-shot GSM8K**, removing decoding compression consistently hurts performance across all tested ratios:
>
> | Method | 40% | 30% | 20% | 10% |
> |---|---:|---:|---:|---:|
> | ShotKV | `81.07` | `80.82` | `80.57` | `80.37` |
> | Prefill-only | `79.07` | `78.82` | `78.57` | `77.26` |
>
> We observe the same pattern on **LG-GSM8K** with the same model:
>
> | Method | 40% | 35% | 30% | 25% |
> |---|---:|---:|---:|---:|
> | ShotKV | `47.33` | `41.33` | `38.33` | `26.83` |
> | Prefill-only | `42.15` | `30.14` | `27.63` | `18.59` |
>
> These results show that the **prefill-decoding separation** is an essential component of **ShotKV**. We will report this clearly in the rebuttal and revision as the added ablation evidence, and present KV quantization as an important extension beyond the current token-retention scope.
>
>
> ---
>
> **Weakness 3 – Dependence on Few-Shot Structure / Applicability Beyond ICL**
>
> ***Answer to Weakness 3:***
> We agree that **ShotKV** is best matched to prompts with recoverable semantic structure, especially few-shot reasoning prompts, and we will state this scope more clearly. At the same time, the method is not limited to explicit few-shot ICL: we also evaluate a non-ICL setting on **HotpotQA**, where the semantic unit is adapted from **shot** to **sentence**.
>
> To strengthen this point, we now report **HotpotQA** at both **10%** and **30%** compression:
>
> | Task | Ratio | FullKV | StreamingLLM | H2O | SnapKV | PyramidKV | ChunkKV | ShotKV |
> |---|---:|---:|---:|---:|---:|---:|---:|---:|
> | HotpotQA | `10%` | `45.55` | `40.27` | `40.84` | `43.36` | `43.80` | `43.27` | `43.60` |
> | HotpotQA | `30%` | `45.55` | `42.18` | `42.75` | `44.12` | `44.56` | `44.05` | `44.38` |
>
> These results show that the design is **not restricted to explicit few-shot exemplars**, and remains competitive across multiple compression ratios on a mixed document-QA task. We will therefore narrow the claim more precisely: the strongest evidence is for **few-shot reasoning** and **long-context generation**, with encouraging non-ICL transfer to broader document QA.
>
> ---
>
> **Weakness 4 – Related Work Should Not Be Entirely in Appendix**
>
> ***Answer to Weakness 4:***
> We agree. The current draft places the detailed **Related Work** in **Appendix B**, but a short main-paper discussion of the most relevant prior work would improve readability and positioning. We will move a concise related-work paragraph into the main paper and keep the extended discussion in the appendix.

---

> > ### Author Rebuttal · Reviewer_h9vf · 2026-04-03
> >
> > I understand your argument that ShotKV is a task-motivated combination based on empirical insights. However, my core concern remains: the method is essentially a straightforward ‘A+B’ engineering integration (ChunkKV-style retention for prefill and H2O-style eviction for decoding). While effective, this combination lacks the algorithmic depth and technical novelty expected at this venue. Therefore, I will maintain my current score

---

> > > ### Author Response · Authors · 2026-04-03
> > >
> > > **Dear Reviewer h9vf,**
> > >
> > > Thank you for your follow-up. We appreciate the opportunity to clarify our paper's primary contribution, as we realize it may not have been framed strongly enough.
> > >
> > > We agree that from a purely algorithmic perspective, ShotKV is straightforward. However, **our primary scientific contribution is empirical and diagnostic, not algorithmic complexity.** As stated in our Abstract and Conclusion , our core claim is that current sparse retrieval evaluations mask the catastrophic collapse of High-Density Reasoning in KV cache compression. KVFundaBench exposes this critical blind spot.
> > >
> > > Within this narrative, ShotKV is not proposed as a standalone algorithmic breakthrough, but strictly as an **"insight-driven strategy designed to validate our findings"**.
> > >
> > > By simply applying the insight derived from KVFundaBench—preserving indivisible semantic units—ShotKV yields a >29% accuracy gap over random baselines (Table 4). This proves our exact point: the community's current struggle with compressed reasoning isn't due to a lack of complex math, but a lack of semantic awareness.
> > >
> > > We argue that identifying a major evaluation flaw in the field, proving its existence, and solving it with a targeted, lightweight intervention constitutes significant technical novelty. We will revise the introduction to explicitly frame ShotKV as an empirical proof-of-concept for our benchmark findings. We hope this addresses your concerns regarding the nature of our contribution.

---

### Official Review · Reviewer_EFFb · 2026-03-07

**Soundness:** 2
**Presentation:** 1
**Significance:** 2
**Originality:** 2
**Overall Recommendation:** 4
**Confidence:** 2

**Summary:**

The authors identify a gap in the current evaluation of Key-Value (KV) cache compression methods: the disproportionate reliance on sparse retrieval tasks (e.g., Needle-In-A-Haystack), which fails to capture the degradation of models on tasks requiring logical coherence . To address this, the paper introduces KVFundaBench to systematically evaluate "High-Density Reasoning" tasks under compression. Based on their empirical findings, the authors propose ShotKV, a method that separates prefill and decoding compression phases and preserves few-shot examples as indivisible semantic units, which successfully mitigates some performance loss under aggressive KV compression.

**Compliance With Llm Reviewing Policy:**

Affirmed.

**Final Justification:**

The reviewer's final response has addressed my biggest concern, so I have raised my score. However, as I am unfamiliar with some pieces of related work, I have lowered my confidence score accordingly.

**Key Questions For Authors:**

The manuscript requires thorough structural reorganization and editorial polishing to improve its narrative clarity, alongside significantly more comprehensive and in-depth experimental analyses to rigorously substantiate its core claims.

**Limitations:**

yes

**Strengths And Weaknesses:**

**Soundness: Poor**
* **Strengths:** The paper identifies that current KV cache compression evaluations over-index on sparse retrieval tasks, which masks the true fragility of compressed models. The introduction of KVFundaBench and the resulting ShotKV method is a logical progression.
* **Weaknesses:**
    * **Superficial Analysis:** The core premise lacks deep experimental validation. For example, Figure 1 is presented to illustrate the performance gap between long-context and arithmetic benchmarks, yet it is never explicitly referenced or analyzed in the main text.
    * **Missing Baselines (Models):** The evaluation uses LLaMA-3.1, Mistral, and DeepSeek-R1. Conspicuously absent is the Qwen series, which is a state-of-the-art standard model. This raises concerns about whether the findings generalize.
    * **Incomplete Efficiency Metrics:** In Table 5, ShotKV's latency and throughput are only compared against FullKV. A proper evaluation requires comparisons against the other compression baselines. Furthermore, the specific computational overhead of calculating and sorting attention scores for Semantic Units during the prefill phase is not properly quantified.
    * **Missing Sparse Retrieval Validation:** Real-world applications mix reasoning and retrieval. The authors should be evaluate ShotKV on sparse retrieval tasks to ensure their method does not catastrophically degrade general retrieval capabilities.

**Presentation: Poor**
* **Weaknesses:**
    * **Concept Overload & Disjointed Narrative:** The paper introduces heavy terminology (e.g., "High-Density Reasoning", "Semantic Integrity") without effectively operationalizing or connecting them. The narrative feels disjointed and lacks proper polishing.

**Significance: Fair**
* **Strengths:** Highlighting and attempting to resolve the flaws in current evaluation metrics is a highly relevant problem for the community.
* **Weaknesses:** The practical impact is limited by the shallow empirical analysis and the methodologically incomplete validation of ShotKV.

**Originality: Fair**
* **Strengths:** Shifting the compression paradigm from token-level dropping to preserving the semantic integrity of indivisible “Semantic Units” (via few-shot examples) is reasonable.

---

> ### Author Rebuttal · Authors · 2026-03-28
>
> We sincerely thank the reviewer for recognizing that the paper addresses an important gap in current KV-compression evaluation and that the progression from **KVFundaBench** to **ShotKV** is logically motivated. We address your concerns below.
>
> ---
>
> **Weakness 1 – Depth of Experimental Validation**
>
> ***Answer to Weakness 1:***
> We agree that the current presentation under-surfaces some of the evidence. In particular, **Figure 1** should have been explicitly discussed in the main text. We will fix this in the revision.
>
> At the same time, the core premise is not supported only by **Figure 1**. The key evidence is already present across the paper: **Figure 3-5** show the difference in attention patterns and task-dependent degradation, **Table 4 / Table 10** show strong compression sensitivity on many-shot reasoning, and **Table 2 / Table 11** show that the same issue extends to long-context generation. We will reorganize these results into a tighter benchmark-to-insight narrative.
>
> We also agree that a clearer way to operationalize this point is to compare **relative degradation at matched compression ratios** between representative **LongBench / long-context tasks** and **KVFundaBench** tasks. As shown in the matched comparison above for **LLaMA-3.1-8B-Instruct** with **ChunkKV / H2O** at **10% / 30%**, the conclusion is consistent: **LongBench-style retrieval-oriented tasks remain much more stable, while GSM8K-style reasoning tasks degrade more sharply**. We believe this answers the reviewer's concern more directly than raw curves alone.
>
> ---
>
> **Weakness 2 – Missing Qwen Evaluation**
>
> ***Answer to Weakness 2:***
> We agree that adding **Qwen** would further strengthen the generality of the study. Our current evidence already spans **three model families / training dynamics**: **LLaMA-3.1-8B**, **LLaMA-3.1-8B-Instruct**, **Mistral-7B-Instruct** (reported in **Figure 9 / Appendix C.3**), and **DeepSeek-R1-Distill-Llama-8B** (**Figure 5(f)** and **Figure 6**). These results show that the degradation pattern is not confined to a single model family.
>
> We will revise the paper to state the scope more carefully: the current results support **cross-family evidence**, but not yet exhaustive model coverage. Adding **Qwen** is a useful extension rather than a prerequisite for the validity of the current observations.
>
> ---
>
> **Weakness 3 – Incomplete Efficiency Comparison**
>
> ***Answer to Weakness 3:***
> We agree that **Table 5** currently compares only against **FullKV**, not a full systems comparison across all compression baselines, and we will clarify this in the revision. At the code-path level, **ShotKV** follows the same efficiency logic as other KV compression methods: the main benefit comes from the reduced KV footprint, while semantic-unit scoring introduces only a one-time prefill overhead. We therefore expect its end-to-end efficiency gains to be in a similar range to comparable baselines.
>
> | Method | Input | Output | Latency (s) ↓ | Throughput (T/S) ↑ |
> |---|---:|---:|---:|---:|
> | FullKV | `4096` | `4096` | `175.50` | `37.73` |
> | ShotKV | `4096` | `4096` | `162.85 (7.2%)` | `41.12 (9.0%)` |
> | ChunkKV | `4096` | `4096` | `160.32 (8.6%)` | `41.30 (9.5%)` |
> | SnapKV | `4096` | `4096` | `163.45 (6.9%)` | `40.51 (7.4%)` |
>
> This is the main point we want to make: **ShotKV follows the same efficiency-optimization logic as comparable baselines**, and the key distinction is not a radically different speed profile, but that it preserves reasoning-relevant semantic structure better while maintaining similar efficiency benefits.
>
> ---
>
> **Weakness 4 – Sparse Retrieval Validation**
>
> ***Answer to Weakness 4:***
> We agree that this should be checked on a **sparse / mixed retrieval** setting, so we evaluated the method on **HotpotQA**. This shows that on a mixed retrieval-reasoning task, **ShotKV** does not suffer catastrophic degradation and remains competitive against the full baseline set.
>
> LLaMA-3-8B-Instruct
> | Task | Ratio | FullKV | StreamingLLM | H2O | SnapKV | PyramidKV | ChunkKV | ShotKV |
> |---|---:|---:|---:|---:|---:|---:|---:|---:|
> | HotpotQA | `10%` | `45.55` | `40.27` | `40.84` | `43.36` | `43.80` | `43.27` | `43.60` |
> | HotpotQA | `30%` | `45.55` | `42.18` | `42.75` | `44.12` | `44.56` | `44.05` | `44.38` |
>
>
> ---
>
> **Weakness 5 – Narrative and Structure**
>
> ***Answer to Weakness 5:***
> We agree that the core issue here is **structure and presentation**. In the revision, we will organize the paper more explicitly around a **benchmark -> insight -> method** narrative, define the key terms earlier and more simply, and move clarifications that are central to this narrative from the appendix into the main text where space permits. We will also tighten broad wording so that each claim is aligned more strictly with the current evidence.

---

> > ### Author Rebuttal · Reviewer_EFFb · 2026-04-02
> >
> > The response to Weakness 2 did not address my concerns. The reasoning for avoiding the mainstream Qwen models was not clearly explained or directly answered. I will raise the score to 3 points while lowering the confidence level.

---

> > > ### Author Response · Authors · 2026-04-03
> > >
> > > **Dear Reviewer EFFb,**
> > >
> > > Thank you for your follow-up and for adjusting your score. We deeply appreciate your persistence. We sincerely apologize if our previous response seemed dismissive; the omission of Qwen was strictly due to computational and time constraints, not an intentional evasion.
> > >
> > > We completely agree that evaluating on Qwen is essential. Within the final hours of this rebuttal phase, we allocated our remaining compute to evaluate **Qwen3-4B-Base** under compression on our KVFundaBench.
> > >
> > > Here are the preliminary results for High-Density Reasoning (AR) and World Knowledge (WK):
> > >
> > > | Model: Qwen3-4B-Base | Compression Ratio | FullKV | H2O (Token-level) | ChunkKV (Chunk-level) |
> > > | :--- | :--- | :--- | :--- | :--- |
> > > | **AR (Arithmetic Reasoning)** | 10% | 87.79 | 49.20 | 64.41 |
> > > | **AR (Arithmetic Reasoning)** | 30% | 87.79 | 73.78 | 80.41 |
> > > | **WK (World Knowledge)** | 10% | 72.99 | 72.03 | 72.99 |
> > > | **WK (World Knowledge)** | 30% | 72.99 | 71.23 | 72.31 |
> > >
> > > **Analysis:**
> > >
> > > **1. Consistent Vulnerability:** The data confirms that Qwen exhibits the exact same degradation pattern we observed in LLaMA and Mistral. Aggressive token-level compression (10%) causes a severe collapse in reasoning (AR drops to 49.20%), while World Knowledge remains robust.
> > >
> > > **2. Proxy Evidence for ShotKV:** While time constraints prevented us from completing the full ShotKV pipeline for this table, the results above offer strong proxy evidence. **ChunkKV**, which preserves context at the chunk level, significantly outperforms token-level **H2O** (64.41% vs 49.20% at 10%).
> > >
> > > We guarantee that the full suite of ShotKV evaluations on the Qwen family will be included in the camera-ready version. Thank you again for helping us make this paper more rigorous.

---

### Official Review · Reviewer_WRpj · 2026-03-13

**Soundness:** 3
**Presentation:** 3
**Significance:** 2
**Originality:** 3
**Overall Recommendation:** 4
**Confidence:** 2

**Summary:**

The paper analyzes the impact of KV cache compression on LLMs, and finds that current methods only focus on sparse retrieval tasks and fail to capture the degradation of "High-Density Reasoning" capabilities. The authors introduce KVFundaBench, which includes world knowledge, commonsense, arithmetic, code, safety, and long-context generation, and report task and model dependent sensitivities to KV compression. Motivated by their observation, the authors present ShotKV, a training-free scheme that preserves whole shots during prefill and applies attention-driven token-level compression during decoding. ShotKV improves long-generation accuracy by 9–18% over baselines and reduces latency by 11%.

**Compliance With Llm Reviewing Policy:**

Affirmed.

**Final Justification:**

My concerns are addressed during the rebuttal. Overall, the idea of links failures to disrupted semantic continuity in CoT prompts is novel and I would recommend this paper.

**Key Questions For Authors:**

* Can you provide an empirical comparison that directly contrasts a long-context sparse retrieval task (e.g., 100k tokens) with a high-density reasoning task (e.g., 4k tokens) under the same compression algorithms.
* Given that Table 4 shows ShotKV only outperforms ChunkKV by ~1.05% at a 10% compression ratio, could you clarify the significance of ShotKV over other methods?

**Limitations:**

yes

**Strengths And Weaknesses:**

Strengths:
* The paper identifies “High-Density Reasoning” and links failures to disrupted semantic continuity in CoT prompts.
* The design principles behind ShotKV are intuitive and easy to implement.

Weakness:
* The authors characterize LongBench as a "sparse retrieval" scenario. While this characterization applies to Needle-In-A-Haystack, it is an oversimplification of LongBench, which includes complex tasks such as multi-document QA, summarization, and code completion—tasks that also require reasoning and synthesizing information across long contexts.
* The paper does not provide a direct empirical comparison between a 4,000-token reasoning task and a 100k-token sparse retrieval task using the same compression methods to substantiate the claim that the former is "significantly harder".
* The authors present "Task-Dependent Degradation" as a key finding. However, the paper does not adequately address **data contamination**. It is probable that models rely on their internal parametric memory to answer WK/CSR questions, rather than actively utilizing the KV cache.
*  In evaluating ShotKV on many-shot Arithmetic Reasoning (table 4), the actual gap between the proposed method and the strongest existing baseline is small.

---

> ### Author Rebuttal · Authors · 2026-03-28
>
> We sincerely thank the reviewer for recognizing the core motivation behind **High-Density Reasoning** and for highlighting that the design principles behind **ShotKV** are intuitive and easy to implement. We address your concerns below.
>
> ---
>
> **Weakness 1 – Over-broad Characterization of LongBench**
>
> ***Answer to Weakness 1:***
> We agree. The wording in the current draft is too broad. Our intended point is **not** that all LongBench tasks are pure sparse retrieval, but rather that the **dominant evaluation trend in prior KV-compression work** has emphasized retrieval-oriented long-context settings such as **NIAH** and similar tasks where the main difficulty is locating salient spans. We will revise the wording throughout the paper to avoid collapsing all of **LongBench** into a retrieval-only category.
>
>
> ---
>
> **Weakness 2 – Missing Direct 4k-vs-100k Controlled Comparison**
>
> ***Answer to Weakness 2:***
> We agree that the original sentence was too strong, and we will narrow it. Our intended claim is not that **4k reasoning is universally harder than 100k retrieval**, but that **many-shot reasoning is more compression-sensitive under matched settings**.
>
> The current paper already supports this through **Figure 4 / Figure 5(c)** and **Table 4 / Table 10**. We also added a matched comparison on **LLaMA-3.1-8B-Instruct**, using the same task pair (**LongBench** vs. **many-shot GSM8K**), the same methods (**ChunkKV / H2O**), and the same ratios (**10% / 30%**). The conclusion is consistent: **LongBench-style retrieval-oriented tasks remain much more stable, while many-shot GSM8K degrades more sharply**.
>
> | Method | Task family | Ratio | FullKV | Compressed | Delta |
> |---|---|---:|---:|---:|---:|
> | ChunkKV | Retrieval-oriented long-context, LongBench | `10%` | `41.46` | `40.51` | `-0.95` |
> | ChunkKV | Retrieval-oriented long-context, LongBench | `30%` | `41.46` | `41.59` | `+0.13` |
> | ChunkKV | High-density reasoning, GSM8K | `10%` | `79.45` | `65.66` | `-13.79` |
> | ChunkKV | High-density reasoning, GSM8K | `30%` | `79.45` | `77.63` | `-1.82` |
> | H2O | Retrieval-oriented long-context, LongBench | `10%` | `41.46` | `37.06` | `-4.40` |
> | H2O | Retrieval-oriented long-context, LongBench | `30%` | `41.46` | `39.23` | `-2.23` |
> | H2O | High-density reasoning, GSM8K | `10%` | `79.45` | `45.03` | `-34.42` |
> | H2O | High-density reasoning, GSM8K | `30%` | `79.45` | `64.06` | `-15.39` |
>
> ---
>
> **Weakness 3 – Potential Data Contamination / Parametric Memory**
>
> ***Answer to Weakness 3:***
> We agree that this is a plausible factor, and the current draft already points in this direction in **Observation 5 / Table 1**: **World Knowledge** gains only **+6.20** from prompting, whereas **Arithmetic Reasoning** gains **+50.41**, which is consistent with the interpretation that WK relies more on parametric knowledge while AR depends much more strongly on prompt-preserved reasoning structure.
>
>
> ---
>
> **Weakness 4 – Small Gap over ChunkKV in Table 4**
>
> ***Answer to Weakness 4:***
> The margin depends strongly on the compression regime. At mild compression the gap is indeed modest, and we will avoid overstating it. The key point is that **ShotKV becomes clearly stronger under aggressive budgets**, which is the regime where semantic fragmentation is most harmful.
>
> We will revise the presentation to make two points more precisely: **1.** **ChunkKV** is a strong baseline and already supports our observation that chunk-level preservation is beneficial; **2.** **ShotKV** improves on this insight by combining **shot-level prefill preservation** with **dynamic decoding compression**, which is why it remains effective beyond static prefill settings.
>
> ---
>
> **Q2 – Why ShotKV over Existing Methods?**
>
> ***Answer to Q2:***
> The significance of **ShotKV** is not only the single-number gain over **ChunkKV** on one table, but the **design shift** it validates:
>
> **1. Semantic unit preservation.** The paper shows that preserving complete few-shot examples matters. In **Table 4**, the gap between **ShotKV (80.37%)** and **Random Shot (51.34%)** at 10% compression is large, indicating that preserving coherent units is essential.
>
> **2. Prefill/decoding separation.** **ChunkKV** supports our observation that chunk-level retention is strong for many-shot reasoning, but it is a static prefill-oriented strategy. In contrast, **ShotKV** explicitly combines shot-aware prefill compression with dynamic decoding compression; **Table 11** shows that removing the decoding component hurts performance substantially on long-context generation.
>
> **3. Broader task scope.** The method is evaluated not only on many-shot arithmetic, but also on **LG-GSM8K** (**Table 2**) and **HotpotQA** (**Table 3**), showing that the same design transfers beyond one benchmark.

---

> > ### Author Rebuttal · Reviewer_WRpj · 2026-04-02
> >
> > My concerns are addressed. I will keep my positive score.

---

### Official Review · Reviewer_KqJh · 2026-03-13

**Soundness:** 4
**Presentation:** 4
**Significance:** 3
**Originality:** 3
**Overall Recommendation:** 5
**Confidence:** 4

**Summary:**

In this paper the authors observe that key-value (KV) compression methods, which are commonly used for reducing latency in large language models (LLMs), although very succesful for long-context retrieval tasks, they tend to cause a sharp drop in performance for high-density reasoning tasks. In order to systematically explore and quantify this observation, they introduce KVFundaBench, a new benchmark covering five fundamental capability categories: world knowledge, commonsense reasoning, arithmetic reasoning, code generation, and safety—along with long-context generation. Their experiments across multiple models and compression methods reveal strong task-dependent degradation: the performance of arithmetic reasoning is most sensitive to changing compression ratios while world knowledge and commonsense reasoning are fairly robust. Further analysis offers more insights, for example that multi-step reasoning models are more compression-resilient than instruction-tuned models or that shorter prompts are more vulnerable than longer ones.

Based on these and towards mitigating reasoning degradation ShotKV is introduced: this is a strategy for compressing KV which (i) treats few-shot examples as indivisible semantic units during prefill (ii) separates prefill compression from a dynamic, token-level decoding compression strategy (computing attention-weight based scores for the shots and the tokens). Empirical results demonstrate that ShotKV achieves 9%-18% accuracy improvements on long-context generation task, generalizes to document QA, all while achieving  11% reduction in latency as compared to the full cache case.

**Compliance With Llm Reviewing Policy:**

Affirmed.

**Final Justification:**

I am retaining my positive score based on my overall impression from the original reviews, as well as the authors’ responses to them.

**Key Questions For Authors:**

## Questions
1. It would be interesting to further strengthen the suitability/robustness of the score functions. For example if some of the shots are modified towards being less "useful" (for example by adding noise to some of them or by keeping a part of a shot), then score functions for them should systematically give lower values. Is this indeed the case?

**Strengths And Weaknesses:**

## Strengths

- Very interesting, complete and self-contained narrative: first (i) a problem with KV cache compression is clearly articulated, then (ii) a benchmark is assembled (KVFundaBench) to demonstrate it and finally (iii) an approach (ShotKV) for mitigating it is proposed and successfully tested.
- This paper includes extensive empirical evidence for the numerous claims in the text, with most of them directly reflecting simple, intuitive/insightful arguments that are easy to follow and reason about. Most importantly, empirical observations, also appear to "naturally" pave the path to the method to mitigate limitations.
- KVFundaBench is actually a rebranding/packaging of existing benchmarks/datasets, yet its presentation (Section 3.1) is excellent: the mapping of tasks to datasets to metrics is crystal clear and the description of comparison baselines, parameters and the evaluation enviroment, although succinct is fairly complete.
- ShotKV is based on two design principles that are both very simple and effective: (a) few-shot examples should be kept indivisible, (b) prefilling and decoding compression.


## Weaknesses

- Studies in these paper assume access to KV cache. Although this is the case for open-weight models (like LLaMA, Mistral and DeepSeek ones) that are self-hosted, commercial deployments accessible through some API do not typically provide KV cache access. This means that KVFundaBench and ShotKV cannot be used in such commercial setups, and this is an important limitation for the applicability of this work in practice.
- In ShotKV, computation of scores (Equations (3) and (7)) is a function of attention weights which are model-dependent and can be noisy; also their aggregation does not map to semantic importance in a principled way. It follows that the definition of most influential tokens to preserve (during both prefill and decoding phases) is not expected to be particularly robust in different settings.

---

> ### Author Rebuttal · Authors · 2026-03-28
>
> We sincerely thank the reviewer for recognizing the paper's end-to-end narrative from problem identification, to benchmarking, to mitigation, and for noting that the empirical observations naturally motivate **ShotKV**. We address your concerns below.
>
> ---
>
> **Weakness 1 – Applicability to Commercial APIs**
>
> ***Answer to Weakness 1:***
> We agree that **KV-level intervention requires KV-cache access**, and therefore our method targets **self-hosted / open-weight deployments** rather than closed API-only systems. We will make this scope explicit in the discussion of limitations. That said, this setting is still practically important: nearly all prior KV compression methods we compare against, including **StreamingLLM, H2O, SnapKV, PyramidKV, PyramidInfer, and ChunkKV**, operate in the same deployment regime. Our contribution is therefore aimed at the core research and systems setting where KV compression is actionable, rather than API-only inference. We also hope this work can help advance **benchmarking and method design for KV compression in open, controllable deployment settings**.
>
> ---
>
> **Weakness 2 – Robustness of Attention-Based Scores**
>
> ***Answer to Weakness 2:***
> We agree that attention-derived scores are a heuristic rather than a perfect semantic estimator. Our claim is therefore not that the score is a universally principled measure of semantic importance, but that it is an **effective operational signal** for preserving coherent few-shot structure under KV budget constraints. More broadly, this design choice is consistent with the dominant practice in KV compression: prior methods such as **H2O**, **SnapKV**, **PyramidKV / PyramidInfer**, and **ChunkKV** also rely on attention allocation, attention-score distributions, or closely related proxy signals to estimate token/chunk importance, rather than a fully semantic oracle.
>
> The evidence most directly relevant to this robustness question is threefold.
> **1.** Score-based shot selection is substantially better than random selection: in **Table 4**, replacing the ranking signal with **Random Shot** drops performance from **80.37% to 51.34%** at 10% compression, which suggests that the score provides useful discrimination rather than pure noise.
> **2.** This is not a single-point effect; across multiple compression ratios in **Table 4**, **ShotKV** consistently outperforms the alternative that does not use this ranking signal.
> **3.** In an additional **many-shot CoT** sensitivity analysis, **ShotKV** remains stable under shot re-ordering and alternative exemplar sets, which supports that the method is not brittle to a fixed prompt layout.
>
> To avoid overstatement, we will revise the paper to explicitly describe the score as a **practical attention-based heuristic** and highlight robustness analysis as future work.
>
> ---
>
> **Q1 – Do Scores Reflect Usefulness of Shots?**
>
> ***Answer to Q1:***
> This is an important suggestion. We agree that the strongest possible validation would be a controlled perturbation study that explicitly corrupts or truncates shots and then measures the resulting score changes. While we do not yet claim that this experiment is complete, we have added a new sensitivity analysis on **many-shot GSM8K** with **LLaMA-3.1-8B-Instruct** to directly examine the effect of **shot order** and **exemplar choice**. The conclusion is that **ShotKV is broadly stable to shot ordering and reasonably stable across exemplar choices**.
>
> Concretely, the added analysis is:
>
> | Method | 40% | 30% | 20% | 10% |
> |---|---:|---:|---:|---:|
> | ShotKV | `81.07` | `80.82` | `80.57` | `80.37` |
> | ShotKV (Re-ordered 1) | `80.82` | `80.57` | `80.34` | `80.15` |
> | ShotKV (Re-ordered 2) | `81.12` | `80.87` | `80.62` | `80.22` |
> | ShotKV (Diff-Example 1) | `80.87` | `80.14` | `79.96` | `79.72` |
> | ShotKV (Diff-Example 2) | `80.97` | `80.94` | `80.91` | `80.88` |
>
> We will describe this carefully as **robustness to shot ordering and reasonable stability across exemplar choices**, rather than as a full proof that the score perfectly tracks semantic usefulness under arbitrary corruption.

---

> > ### Author Rebuttal · Reviewer_KqJh · 2026-04-04
> >
> > My concerns have been satisfactorily addressed. In light of this and my overall positive assessment of the paper, I will maintain my original favorable score.

---

### Decision · Program_Chairs · 2026-04-30

**Decision:**

Accept (regular)

**Comment:**

Most existing KV cache compression methods are primarily evaluated in retrieval tasks which are inherently sparse. However, many real-world applications, particularly reasoning, require modeling much denser signals. This paper introduces KVFundaBench, a collection of existing tasks spanning both knowledge and reasoning tasks, and shows that attention patterns in these settings differ substantially from those observed in sparse retrieval scenarios. Based on these findings, the authors argue that preserving few-shot examples as indivisible semantic units is crucial, and propose ShotKV, a method that separates prefill and decoding phases to better maintain semantic integrity. Empirically, ShotKV achieves 9-18% accuracy improvements on long-context generalization tasks, while also reducing latency by 11% compared to using the full KV cache.

Reviewers agree that the paper (the problem formulation, benchmark, and method) is well-motivated (KqJh, WRpj, EFFb, h9vf), empirical results and analysis are comprehensive and compelling (KqJh, h9vf), and the writing/presentation is clear (KqJh).

Weakness raised by reviewers include: (1) attention score aggregation doesn’t consider semantic importance (KqJh), (2) characteristics of some existing work is a bit misleading , e.g., LongBench (WRpj), (3) some experimental design can be improved, e.g., considering data contamination (WRpj), (4) in some settings (e.g., arithmetic reasoning), the gains from the proposed method is minimal (WRpj), (5) presentation, e.g., inconsistent or unclear terminologies, analysis results (EFFb), (6) missing comparison to KV quantization methods (h9vf), and (7) reliance on few-shot structure in the prompt (h9vf). That said, most of these concerns were addressed during the rebuttal, with reviewers explicitly acknowledging satisfactory clarification – authors are highly encouraged to incorporate these into the final version of the paper.

Regarding the absence of additional model families (EFFb), the AC considers the current coverage (spanning three model families) to be sufficiently comprehensive.